# Causal relationships and predictability of the summer East Atlantic teleconnection

Julianna Carvalho-Oliveira[1,2,3,a], Giorgia Di Capua[4,5], Leonard F. Borchert[1], Reik V. Donner[5,4], and Johanna Baehr[1]

[1]Institute of Oceanography, Center for Earth System Research and Sustainability (CEN), Universität Hamburg, Hamburg, Germany
[2]International Max Planck Research School on Earth System Modelling, Max Planck Institute for Meteorology, Hamburg, Germany
[3]Helmholtz-Zentrum Hereon, Institute of Coastal Systems - Analysis and Modeling, Geesthacht, Germany
[4]Potsdam Institute for Climate Impact Research, Member of the Leibniz Association, Potsdam, Germany
[5]Department of Water, Environment, Construction and Safety, Magdeburg-Stendal University of Applied Sciences, Magdeburg, Germany
[a]present address: Leipzig Institute for Meteorology, Universität Leipzig, Leipzig, Germany
**Correspondence:** Julianna Carvalho-Oliveira (julianna.carvalho_oliveira@uni-leipzig.de)

**Abstract.**

We apply Causal Effect Networks to evaluate the influence of spring North Atlantic extratropical sea surface temperatures (NA-SST) on the summer East Atlantic Pattern (EA) seasonal predictability during the period of 1908-2008. In the ERA-20C reanalysis, we find that the causal link from the meridional NA-SST gradient in spring (expressed by a meridional "SST index") to the summer EA is robust during the period from 1958 to 2008, with an estimated causal effect expressed by a $\beta$-coefficient of about 0.2 (a 1 standard deviation change in the spring SST index causes a 0.2 standard deviation change in the EA 4 months later). However, this causal link is not evident when analysing the entire period from 1908 to 2008. When performing the analysis on 45-year-long time series randomly sampled from this late period, we find the strength of the causal link to be affected by interannual variability, suggesting a potential modulation by an external physical mechanism. In addition to the summer EA, we find that the spring SST index has an estimated causal effect of about -0.2 on summer 2-metre air temperatures over northwestern Europe. We then use different datasets from the MPI-ESM-MR to analyse the 1908-2008 period, focusing on a historical simulation and a 30-member initialised seasonal prediction ensemble. We specifically test the model's ability to reproduce the causal links detected in ERA-20C and evaluate their impact on the model's predictive skill for European summer climate. We find that MPI-ESM-MR generally fails to reproduce the causal link between the spring SST index and the summer EA across the datasets. The 30-member initialised ensemble occasionally reproduces the causal link, though it typically underestimates its strength. We perform a predictive skill assessment conditioned on the spring SST index causal links for July-August sea level pressure, 500 hPa geopotential height and 2-metre air temperatures for predictions initialised in May. Our results suggest that while the overall impact may be limited, leveraging these causal links locally could help to constrain and improve the seasonal prediction skill of European summer climate.

# 1 Introduction

The summer East Atlantic Pattern (EA) is an important atmospheric teleconnections influencing weather and climate in the Euro-Atlantic region (e.g. Comas-Bru and McDermott (2014); Bastos et al. (2016)). Along with the summer North Atlantic Oscillation (NAO), these teleconnections are often used to describe the combined changes in latitude and speed of the North Atlantic jet stream (Woollings et al., 2010) – one of the major modulators of mid-latitude weather extremes (e.g. Rousi et al. (2022)). Understanding the predictability associated with these teleconnections is therefore of paramount importance. Although several recent studies have focused on predictability of the NAO (Domeisen et al., 2018; O'Reilly et al., 2019; Athanasiadis et al., 2020; Klavans et al., 2021), the EA has received less attention. Here, we apply the Peter and Clark momentary conditional independence (PCMCI) causal discovery algorithm to evaluate the influence of North Atlantic extratropical surface temperatures (NA-SST) on the predictability of EA at seasonal timescales.

The most common description of the EA pattern features a well-defined sea level pressure (SLP) centre of action south of Iceland and west of the British Isles, usually defined as the second leading empirical orthogonal function (EOF) of SLP in the Euro-Atlantic region (e.g. Moore et al. (2013)). Wallace and Gutzler (1981) define a positive phase of the EA as characterised by the centre of action exhibiting anticyclonic conditions, featuring the northward extension of the Azores High. A positive EA has been associated with below-average surface temperatures (Cassou et al., 2005; Comas-Bru and Hernández, 2018) and dry spells in parts of Europe (Rousi et al., 2021). Conversely, anomalous cyclonic conditions offshore of Ireland have been suggested to influence heatwaves in Europe for a negative EA phase (e.g. Duchez et al. (2016)). Using a clustering approach (e.g. Cassou et al. (2004); Carvalho-Oliveira et al. (2022)), a positive EA phase is reminiscent of an Atlantic Ridge, whereas a negative EA phase resembles the Atlantic Low. A common feature amongst the different EA definitions is that its centre of action is positioned along the NAO nodal line, thus ultimately modulating the location and strength of the NAO dipole and the North Atlantic storm track (Woollings et al., 2010). That is, summer climate predictability in the Euro-Atlantic region is closely linked to EA variability.

While there is no consensus on the physical processes driving the EA, spring NA-SST have been proposed to influence EA variability and predictability. Gastineau and Frankignoul (2015) suggested that summer 500-hPa geopotential height anomalies in the Euro-Atlantic significantly co-vary with a spring NA-SST tripole pattern in observations over the 20th century. Moreover, Carvalho-Oliveira et al. (2022) suggested that spring North Atlantic SSTs can influence predictive skill of summers dominated by EA in initialised simulations. Based on linear regression analyses of the period 1979–2017, Ossó et al. (2018) and Ossó et al. (2020) proposed a physical mechanism whereby anomalous extratropical North Atlantic SSTs in spring may persist into summer and influence shifts in the eddy-driven jet stream, imprinting at the surface an SLP pattern that resembles the EA. These studies suggest that this mechanism is forced by changes in baroclinicity of the lower troposphere associated with a strong meridional NA-SST gradient in spring located between subpolar and subtropical gyres. The authors hypothesised that the delayed atmospheric response in summer, and not in spring, could be explained by the seasonal evolution of both NA-SST gradient and jet stream position, modulated by a positive coupled ocean–atmosphere feedback that operates primarily in summer.

Nevertheless, while the linear regression-based analysis provided in Ossó et al. (2018) suggests a contribution of spring NA-SST to summer SLP variability, this approach does not imply causation. Disentangling the complex causal-effect pathways underlying the mechanism proposed in Ossó et al. (2020) over a long observational record is a crucial step in evaluating EA predictability in dynamical climate models. Although dynamical seasonal forecasts of European summer climate typically show limited skill (e.g., Mishra et al. (2019)), recent studies suggest that improving the representation of teleconnections can increase forecast skill (Oliveira et al., 2020; Carvalho-Oliveira et al., 2022; Schuhen et al., 2022). The physical mechanism connecting NA-SST variability and jet stream dynamics proposed in Ossó et al. (2020) provides a framework for assessing the broader influence of NA-SST on seasonal predictability of the EA, which is the aim of the present study.

In this paper, we use a Causal Effect Network based on PCMCI (hereafter CEN, Kretschmer et al. (2016)) to test the hypothesis that spring NA-SST causally drives a response in summer SLP and temperature fields in the Euro-Atlantic sector during the 20th century. CEN overcomes spurious correlations caused by autocorrelation, indirect effects, or common drivers (Runge et al., 2014, 2019). It has been successfully applied to hypothesis testing for other tropical and mid-latitude teleconnections in the Atlantic-Pacific region (e.g., Karmouche et al. (2023)), the Indian Ocean (e.g., Di Capua et al. (2020a)), and the Arctic region (e.g., Siew et al. (2020); Kretschmer et al. (2020)).

Specifically, we use CEN to investigate the circumstances under which spring extratropical North Atlantic SSTs causally influence summer EA conditions and their associated impact on surface climate. We also analyse pre-industrial, historical, and initialised simulations with the Max Planck Institute Earth System Model in its mixed-resolution setup (MPI-ESM-MR, Dobrynin et al. (2018)) to evaluate model performance in reproducing the observed NA-SST–EA link, aiming to identify how this performance may constrain the seasonal prediction skill of European summer climate.

## 2 Methodology

### 2.1 Reanalysis and model data

We investigate the NA-SST - EA link first using ERA-20C reanalysis (Poli et al., 2016), and then using model simulations with MPI-ESM-MR (Dobrynin et al., 2018)). The physical variables analysed are NA-SST, SLP and air temperature at 2 metre height (T2m). We use monthly means for each variable as we are testing mechanisms which are expected to act on monthly timescales.

In MPI-ESM-MR, the atmospheric component ECHAM6 (Stevens et al., 2013) has a resolution of T63L95, with a nominal horizontal resolution of 200 km (1.875°) and 95 vertical layers up to 0.01 hPa. The oceanic component MPI-OM (Jungclaus et al., 2013) is coupled to ECHAM6 and has a resolution of TP04L40, with an approximate horizontal resolution of 40 km (0.4°) and 40 vertical layers. External forcing is taken from CMIP5 (Giorgetta et al., 2013).

We investigate how MPI-ESM-MR performs in reproducing the NA-SST - EA link among three independent sets of MPI-ESM-MR simulations. The datasets comprise a pre-industrial control run (piControl), a historical run, and a 30-member seasonal initialised hindcast ensemble (MR-30). Comparing the performance of each set against reanalysis enables us to distin-

guish the role of forcing (from piControl to historical), and of assimilation (historical to initialised ensemble) on the model skill.

The pre-industrial coupled atmosphere/ocean control run piControl has a total length of 1000 years (period 1850-2849) (Giorgetta et al., 2012), with forcing constant in time: orbital parameters and greenhouse gases concentration are fixed at 1850 values; spectral solar irradiance remains constant as the solar cycle average over 1844-1856, and monthly ozone concentrations are fixed at the 11-year average over 1850-1860 (Mauritsen et al., 2012). The historical simulations run from 1850 to 2005 under natural and anthropogenic forcing following CMIP5 protocol (Dobrynin et al., 2018).

Lastly, the hindcast ensemble MR-30 is initialised on 1st of May every year from 1902-2008, with initial conditions taken from an assimilation experiment (Oliveira et al., 2020). In the assimilation experiment, Newtonian relaxation (*nudging*) is used in full-field mode towards all atmospheric and ocean levels except in the boundary layer. The atmosphere conditions of vorticity, divergence, three-dimensional temperature and two-dimensional pressure are assimilated with ERA-20C data. In the ocean, three-dimensional daily mean salinity and temperature anomalies are nudged at a relaxation time of approximately 10 days. To help reduce initialisation shock, the ocean state is derived from an ocean-only simulation performed with MPI-OM forced with the atmospheric variables from ERA-20C, thus maintaining consistency in model physics. The three-dimensional atmospheric and ocean fields of the assimilation experiment form the initial conditions, from which 30 ensemble members are generated by perturbing the atmospheric state with slightly disturbed diffusion coefficients in the uppermost layer.

We focus our analysis on the 101-year period spanning 1908-2008, using data from both the historical simulation, the MR-30 hindcast ensemble, and ERA-20C. In addition, the piControl simulation, with its fixed external forcings, offers a unique opportunity to study long-term internal climate variability free from anthropogenic influences. To capture the full range of natural variability, we leverage the entire 1000-year period of the piControl in our analysis. This approach allows for benchmarking internal variability across observed, historical, and hindcast datasets.

## 2.2 Data-processing and climate indices

We compute anomalies at every gridpoint by removing mean seasonal cycle and linear trend, satisfying data input requirements for the CEN algorithm (Kretschmer et al., 2016). We analyse bimonthly means in March-April (MA) and April-May (AM) for spring NA-SST and July-August (JA) SLP and T2m. We choose to investigate both MA and MA spring windows to allow comparison with previous studies (e.g. Ossó et al. (2018)). In MR-30, we use the assimilation experiment to obtain spring NA-SST fields, and the hindcast ensemble at lead times of 3-4 months to obtain summer SLP, T2m and 500 hPa geopotential height (Z500). We apply area-weighting by multiplying each value with the cosine of its latitudinal location to take into account the dependence of the gridpoint density on latitude.

We calculate the EA index to analyse the summer EA teleconnection. As a first step, we define a reference EA index as the second principal component (PC) of the EOF of JA anomalies of SLP over the Euro-Atlantic sector 70°W-40°E, 25°-80°N calculated from the ERA-20C reanalysis data (e.g. Comas-Bru and McDermott (2014)). Next, EA index values in the model simulations from MPI-ESM-MR are calculated by projecting each ensemble member onto the EA reference EOF pattern. We

consider a positive phase of the EA index when characterised by a centre of positive SLP anomalies that lies south of Iceland and west of the British Isles (e.g. Wallace and Gutzler (1981); Comas-Bru and McDermott (2014), Fig.1a).

We further test the influence of spring extratropical North Atlantic SSTs on the summer EA using the SST index proposed in Ossó et al. (2018). As a second step, we calculate the SST index by subtracting the average NA-SST anomalies over the eastern box (35°W-20°W, 35°-42°N) from the average NA-SST anomalies over the western box (52°W-40°W, 42°-52°N), represented by green boxes in Fig.1b.

To comprehensively investigate the influence of spring NA-SST on summer SLP variability, we incorporate the SLP index introduced by Ossó et al. (2018) alongside the EA index. This approach aims to address the broader significance of pressure dynamics in the region, particularly in relation to the physical mechanism proposed by Ossó et al. (2020). The SLP index is calculated as JA SLP anomalies averaged over the region 45°N-55°N; 25°W-5°W indicated by a blue box in Fig1b. Next, we analyse the impact of NA-SST on summer T2m using two additional indices, i.e. $T2m_{CE}$ and $T2m_{Ridge}$ (Sects.3.2,3.4). The $T2m_{CE}$ index is calculated as JA T2m anomalies averaged over the region 46°N-55°N; 11°E-34°E (indicated by a red box in Fig.A1i), and the $T2m_{Ridge}$ index is calculated over the region 40°N-55°N; 15°W-34°W (indicated by a black box in Fig.7b).

All climate indices are standardised to have mean of zero and standard deviation (SD) of 1 to allow for comparison. Using the aforementioned climate indices, we perform linear regressions and correlations to analyse the linear relationship between the predictor spring NA-SST and the target variables summer EA, SLP index, and $T2m_{CE}$. We use a two-tailed Student's t-test to calculate the statistical significance of point-wise correlations maps. We provide a description of the indices used in the analysis in Table 1.

**Table 1.** Summary of the indices used in our analysis.

| Index | Variable name | Region used for calculation |
|---|---|---|
| $SST_{ind}$ | Sea surface temperature index | Eastern box (35°W-20°W, 35°-42°N); Western box (52°W-40°W, 42°-52°N) |
| $SLP_{ind}$ | Sea level pressure index | 45°N-55°N; 25°W-5°W |
| EA | East Atlantic Pattern | 70°W-40°E, 25°-80°N |
| $T2m_{CE}$ | Air temperature at 2m height for central Europe | 46°N-55°N; 11°E-34°E |
| $T2m_{Ridge}$ | Air temperature at 2m height for Atlantic Ridge region | 44°N-55°N; 15°W-34°W |

## 2.3 Causal Effect Networks

We employ the Causal Effect Network (CEN) method (Runge et al., 2015; Kretschmer et al., 2016) to analyse the causal influence of the spring SST index on summer EA and temperature variability in the Euro-Atlantic sector. CEN allow to represent the output of the Peter and Clark momentary conditional independence (PCMCI) causal discovery algorithm (Runge et al., 2019; Spirtes et al., 2000). We specifically use the PCMCI version 4.2 from the Python package Tigramite (https://github.com/jakobrunge/tigramite). This algorithm is based on iterative conditional independence testing amongst a set of time series

(actors) to assess whether a link between a potential precursor and a target variable at a certain time lag is: *i)* considered spurious, i.e. can be explained by the linear combination of other time series at different lags (conditional independence); or *ii)* considered causal, i.e. cannot be explained by the combined influence of other investigated variables (conditional dependence). In the algorithm, this testing is performed for a minimum and maximum time lag, denoted $\tau_{min}$ and $\tau_{max}$.

We emphasise that the term "causal" should be interpreted cautiously within the context of this study. When we refer to causality, we mean causality relative to the set of investigated variables and under the specific assumptions considered in the PCMCI algorithm (such as the stationarity of time series data). As a consequence, the possibility of remaining spurious correlations cannot be entirely ruled out. The choices of variables included in the analysis is another crucial aspect for determining the causality of the identified links. Yet, this poses a challenge as including more variables enhances the credibility of causal discoveries but introduces complexities. For instance, accommodating numerous variables and significant time lags to address physical delays, like identifying atmospheric teleconnections, leads to high dimensionality. This, in turn, can significantly affect the reliability of statistical outcomes. Hence, a successful application of CEN requires (such as for any data-driven method), expert knowledge of the underlying physical processes, including relevant variables, time-scales and temporal resolution. For a more detailed understanding of the CEN analysis and the PCMCI algorithm, we refer the reader to Runge (2018), which provides a comprehensive description of these techniques.

We visualise the output of PCMCI in a CEN, i.e. a causal graph where nodes represent the investigated variables, arrows indicate the direction of the causal links, and colours denote the strength of these links. The strength is expressed by the standardised linear regression coefficient, denoted $\beta$-coefficient, and defined as the expected change of $Y_t$ in units of its SD induced by raising $X_{t-\tau}$ by 1 SD, while keeping all other potential precursors constant. Moreover, CEN analysis outputs the autocorrelation path coefficient, which represents the causal influence of a variable on itself, as opposed to the Pearson autocorrelation.

We apply causal maps (Di Capua et al., 2020b) to investigate the causal effects of a specific variable on a given atmospheric field along latitude, longitude and time dimensions. This tool builds upon the PCMCI algorithm and CEN approach, and provides a powerful visualisation of spatial patterns. Causal maps display $\beta$-coefficients calculated with the time series of a potential precursor and each grid point of a target atmospheric field. We refer the reader to Di Capua et al. (2020b) for a detailed explanation of this method.

Lastly, the PCMCI parameters are chosen as follows: pc alpha = 0.2, alpha level to print results = 0.1, independence test = ParCorr, significance = 'analytic', masking type 'y'. ParCorr was chosen for its effectiveness in detecting linear relationships, computational efficiency, and established use in related studies (e.g. Siew et al. (2020)), offering clear insights into conditional relationships. Our CEN analysis focuses on $\tau_{min}$ = 3 months and $\tau_{max}$ = 4 months, which for simplicity we refer to as 3 and 4-month lags.

## 2.4 Cross-validation and ensemble subsampling

We perform cross-validation and ensemble subsampling (e.g. Dobrynin et al. (2018)) to investigate the sensitivity of the causal links to data sampling and to better understand the differences in the strength of causal links between ERA-20C and MR-30.

When analysing the period 1958-2008 using observations in Sect. 3.3, we conduct a leave-k-out cross-validation, where we randomly exclude 6 years (approximately 12% of the period) at each iteration. This approach yields 500 different samples of 45-year-long time series, each analysed using CEN with the same hyperparameters (see Sect.2.3). This method allows us to test the robustness of the causal graph structure by assessing how consistent the identified links are across various subsets of the observational data.

We apply a similar cross-validation approach to MR-30, incorporating an additional ensemble subsampling step. Specifically, we first randomly exclude 6 years from the analysed period. Then, for each remaining year, we perform bootstrapping without replacement to randomly select one ensemble member from the 30-member set. This results in a total of 1000 samples of 45-year-long time series. In the CEN analysis, we impose the set of causal parents identified as significant in ERA-20C onto MR-30 and calculate the corresponding causal effects. This allows for a fairer comparison of the strength of the $\beta$-coefficients between the model and observations.

It is important to note that while reducing the length of the time series increases variability and lowers the statistical significance of the $\beta$-coefficients, it does not necessarily diminish the strength of the causal effects themselves. By employing cross-validation and ensemble subsampling, we ensure that our findings are robust to data sampling and sensitive to both the observational record and model ensemble variability.

## 2.5 Predictive skill assessment

In Sect.3.4, we perform a predictive skill assessment for SLP, T2m and Z500 at lead times of 3-4 months in MR-30 against ERA-20C. For this assessment we use point-wise detrended anomaly correlation coefficient (ACC, Collins (2002)). We are interested in assessing the predictive skill conditioned to the strength of significant $\beta$-coefficients (p-value < 0.1). Our hypothesis is that the predictive skill in summer is likely to increase in cases where MR-30 is able to capture the causal link between spring SST index and summer EA, as opposed to cases where the model fails to capture the observed causal link. We refer to these time series as *MR-30 bootstrap ensemble*. For example, we shall assume that we are interested in calculating the conditioned predictive skill of JA Z500. To accomplish this task, we first identify the specific years and ensemble members that correspond to significant $\beta$-coefficients for the spring SST and summer EA. With this information, we can then sample JA Z500 to create a time series of similar length. In case more than one ensemble member is randomly selected in a given year, we calculate an ensemble mean. We then determine the ACC between the MR-30 bootstrap and ERA-20C.

## 3 Results

### 3.1 Characteristics of the observed link: temporal and spatial variability

The spatial pattern of the summer EA in its positive phase is characterised by large-scale cyclonic conditions across the Euro-Atlantic region, except at the anticyclonic centre of action located south of Iceland and west of the British Isles (Fig.1a). A typical surface climate imprint of the summer EA in positive phase correlates with below-average temperatures in continental

Europe (Fig.1c) and below-average precipitation in the British Isles and northwestern Europe (Fig.1d). As explained in Sect2.2, we evaluate spring extratropical North Atlantic SSTs via the SST index, following Ossó et al. (2018). A Pearson correlation

analysis reveals a time-dependent relationship between the AM SST index and the EA in summer (Fig.1e). Over a span of 101 years (1908-2008), this relationship appears weak (r = 0.22, p < 0.05). However, examining the most recent 51 years (1958-2008) shows a doubling of correlation values (r = 0.43, p < 0.05). Furthermore, focusing on the latest 30 years (the period analysed in Osó et al. (2018)) results in correlation values increasing even further to 0.60 (p < 0.05) . The temporal variability of this relationship is well illustrated for correlations calculated using a 20-year running window, which shows a reversal in the

sign of correlations starting from 1945, and highlights an increase in the strength beyond 1958 (Fig.1f). This analysis suggests that the spring SST index - summer EA relationship is nonstationary. Hence, we distinguish the following three periods to scope the remaining analysis: i) *early period*: 1908 - 1957; ii) *late period*: 1958-2008, and iii) *full period*: 1908-2008.

We assess the spatial features of the SST index influence on the summer atmospheric circulation in the different periods to further explore the variability of the spring NA-SST - summer EA relationship. Correlation maps in Fig.A1a-f show distinct

patterns in early and late periods. We find significant correlations between the precursor SST index and summer SLP over a region in the North Atlantic which reasonably coincides with the location of the EA teleconnection centre of action during the late period (Fig.A1b, Fig.1a). The location of this region seems to oscillate about 45°N, remaining south of this latitude in the early period (Fig.A1a), while located northwards in the late one (Fig.A1b). Surrounding this high correlation region, the sign of correlations is opposite between early (Fig.A1a) and late (Fig.A1b) periods. We find similar results using March-April (MA)

NA-SST means, only in weaker strength (e.g. Appendix A Fig. 1).

Regression maps further suggest that the spring SST index is associated with summer SST anomalies, which then influence atmospheric circulation (Fig.A1d-f). Positive values of the AM SST index in spring are associated with warm summer anomalies east of Newfoundland and cool anomalies west off Iberia, leading to concomitant anticyclonic conditions in the ocean located south of Greenland. In the late period (Fig.A1e), these anticyclonic conditions coincide specifically with the position

of the EA centre of action, whereas this association is absent in the early period.

Moreover, we test whether the SST index influences JA T2m via the EA. We find significant correlations between the AM SST index and JA T2m, showing a similar pattern of significant positive correlations west of the British Isles, as in Fig.1c corresponding to JA EA - T2m. We find that correlations between AM SST index and JA T2m show distinct patterns between early and late periods (Fig.A1g,h). A positive phase of the SST index in spring precedes a positive phase of the

235 summer EA (e.g. Fig.A1e), which in turn can be associated with below-average temperatures, primarily over central Europe. To further investigate this relationship, we calculate a T2m$_{CE}$ index, defined as the average summer T2m over the central European region 46°N-55°N; 11°E-34°E, represented by the red box in Fig.A1i. In summary, this analysis reveals that spring extratropical oceanic forcing of the summer atmospheric circulation has a marked temporal and spatial variability over the 20th century, only projecting onto the EA pattern over the late period. This variability might pose a constraint on the predictive skill

of European summer climate based on spring extratropical NA-SST during certain periods of time.

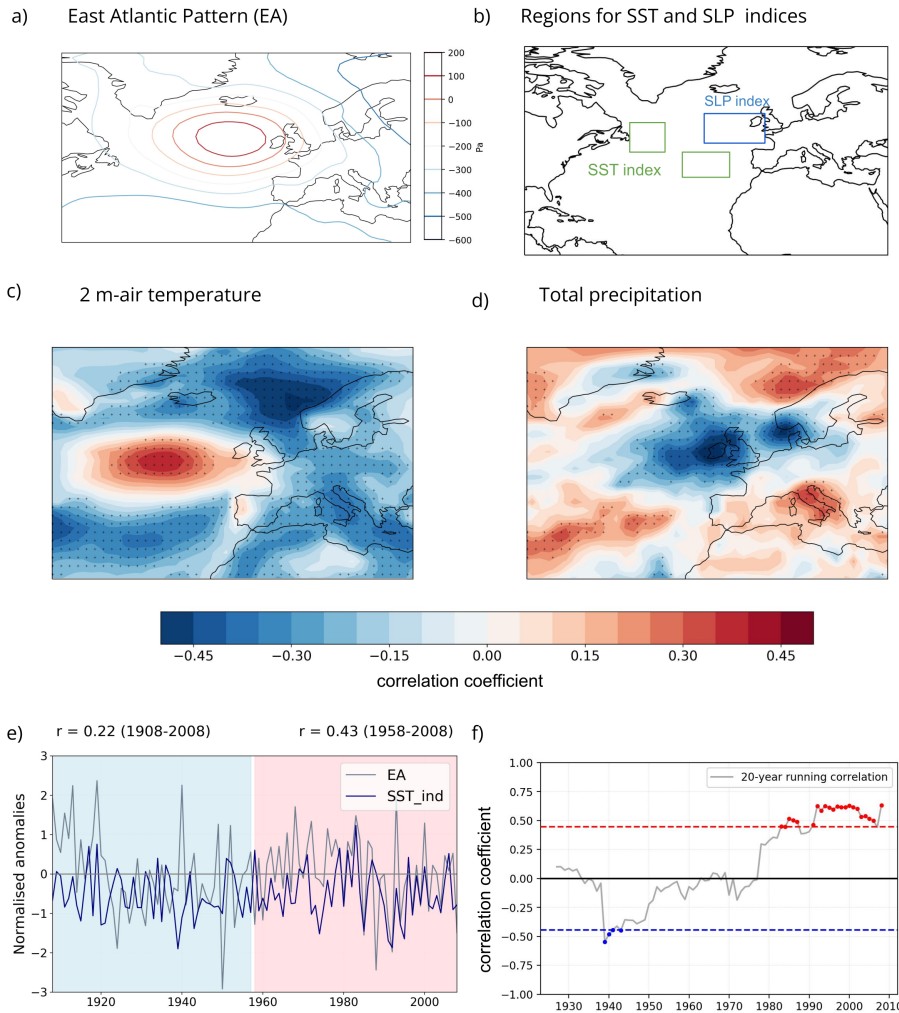

**Figure 1.** Variability and linear relationships of EA in ERA-20C. a) Positive phase of the EA teleconnection, defined as the second EOF of July-August (JA) SLP. b) Regions used to calculate the NA-SST and SLP indices proposed in Ossó et al. (2018). c) Pointwise correlation of EA index with concurrent JA anomalies of 2-metre air temperatures in the full period (1908-2008). d) Same as c), for JA anomalies of total precipitation. e) Time series of April-May (AM) SST (blue) and JA EA (grey) indices in ERA-20C for 1908-2008, smoothed by a 3-year running mean. f) Running-correlation between AM SST and JA EA indices for a 20-year window. Coloured markers indicate significant correlations at the 95% confidence interval, illustrated by dashed lines.

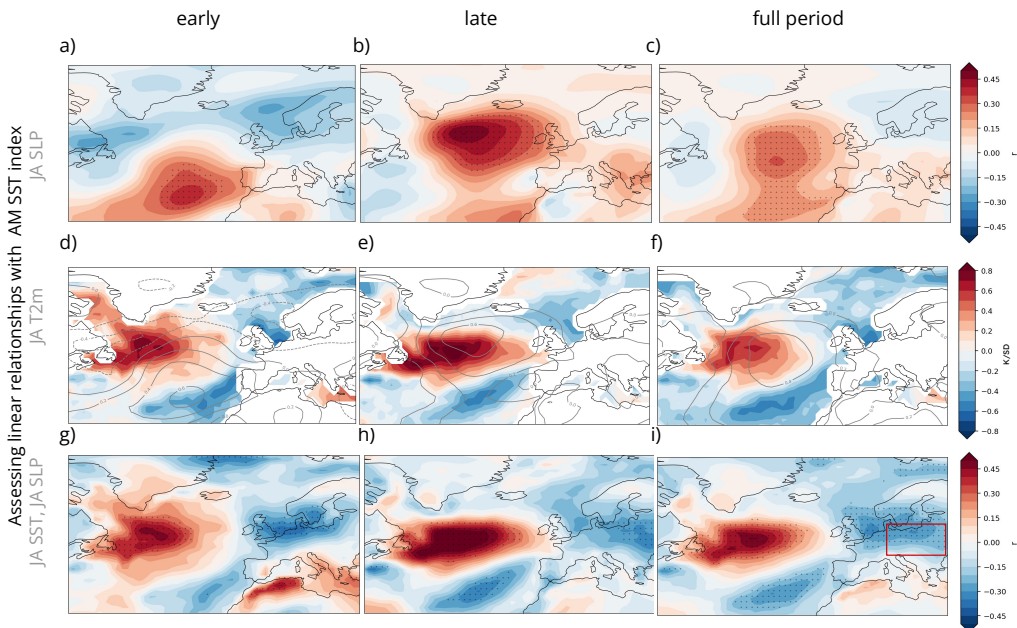

**Figure 2.** Distinct spatial characteristics of the spring SST influence on the summer circulation over the 20th century (for ERA-20C) for *early* (1908-1957, left), *late* (1958-2008, middle) and *full* periods (1908-2008, right column). Top row (a-c) shows point-wise correlation coefficients for the April-May SST index and July-August SLP. Middle row (d-f) shows linear regression maps of July-August NA-SST anomalies (shading) and SLP (contours) against the precursor SST index (normalised by the SD). Contour interval is 0.2 hPaSD$^{-1}$. Bottom row (g-i) shows point-wise correlation coefficients for the April-May SST index and July-August air temperature at 2 metre height. Stippling indicates correlations significant at the 95% confidence level, calculated with a Student's t-test. Box in Fig.2i illustrates the region used to calculate the T2m$_{CE}$ index, as described in the text.

## 3.2 Investigating causality

To further test the robustness of the SST-EA relationship in ERA-20C, we evaluate whether spring SST index and summer EA are conditionally dependent. Specifically, we test the hypothesis that spring SST index is a causal driver for the summer EA, thus excluding autocorrelation effects or common drivers which could lead to spurious links.

First, we build one CEN for each of the three investigated periods in ERA-20C, i.e. early, late and full periods, as defined in Sect.3.1. Besides the EA and SST indices, we include two additional indices in the CEN. The first is the SLP index, defined in Ossó et al. (2018) and illustrated by the blue box in Fig.1b. Thus, we test whether differences between early and late periods (Sec.3.1) are reflected in distinct timing or strength among the EA and SLP indices with SST. The second index concerns summer air temperatures averaged over the region represented by the red box in Fig.A1i (T2m$_{CE}$), which shows significant
anticorrelations with SST. We test whether the spring SST index causally drives changes in summer T2m over central Europe and under which circumstances this holds true.

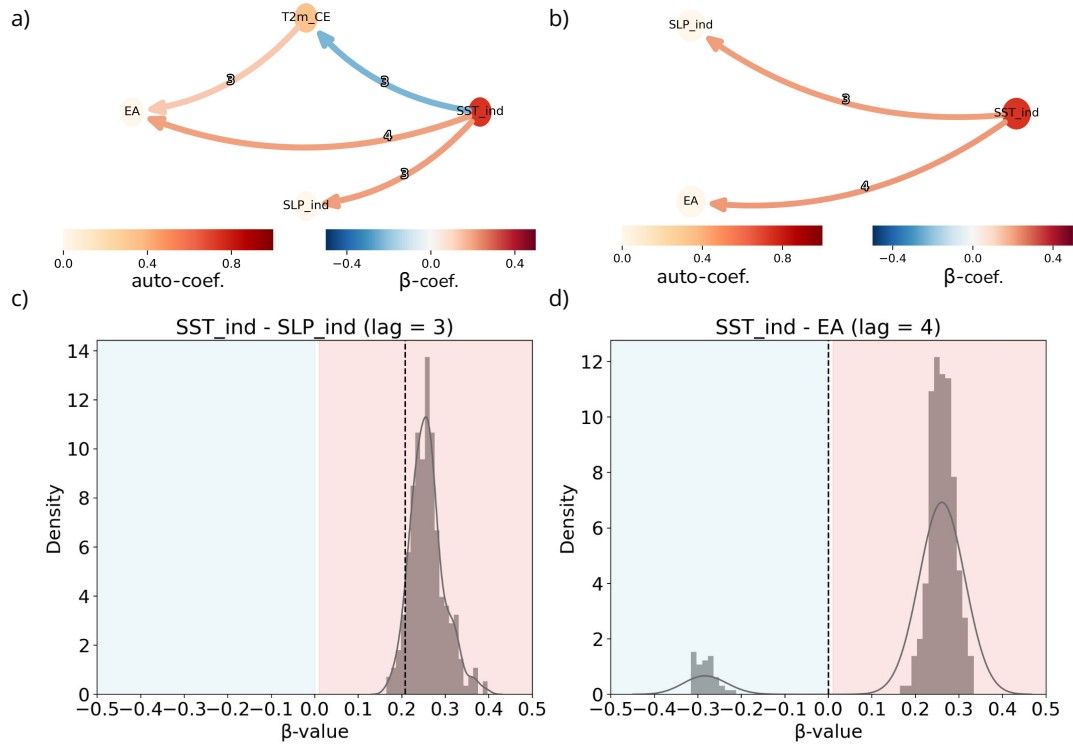

**Figure 3.** Causal Effect Network analysis for the late period (1958-2008) in ERA-20C. Causal graphs between a) SST index, EA teleconnection, SLP index and T2m$_{CE}$ and b) SST index, EA teleconnection and SLP index only. The strength and direction of the causal links is given by the $\beta$-coefficient and is represented by the arrows, whereas the auto-correlation path coefficient is represented for each variable by the respective circle colour. The numbers over each arrow represent the time lag (in months) when the strongest causal link between each variable pair is detected. c-d) Sensitivity of the causal links shown as the PDF of $\beta$-coefficients calculated for a random sample selection of 45 years, iterated 500 times, between the variables: SST and SLP indices at lag 3 (c), and SST index and EA at lag 4 (d). Only causal links with p-value < 0.1 are shown in (a) and (b). Red lines show the correspondent $\beta$-coefficients represented in (a).

Over the late period, we confirm that the spring SST index is a causal driver for both the summer EA and the summer SLP index, at distinct time lags (Fig.3a). The strength of the causal link is expressed by the standardised regression coefficient, denoted $\beta$-coefficient in CEN. At a 4-month lag, we find $\beta_{\text{SST}_{\text{ind}}\rightarrow\text{EA}} \approx 0.22$, which means that a change of 1 standard deviation
(SD) in the MA SST index leads to a change of 0.22 SD in JA EA. We find a causal link of similar strength at a 3-month lag, $\beta_{\text{SST}_{\text{ind}}\rightarrow\text{SLP}_{\text{ind}}} \approx 0.21$ , between AM SST index and JA SLP index, as well as $\beta_{\text{SST}_{\text{ind}}\rightarrow\text{T2m}_{CE}} \approx -0.2$ between AM SST index and JA T2m$_{CE}$. Using the path-tracing rule (e.g. Kretschmer et al. (2021)) we find that about a third of the influence from AM SST index on JA T2m$_{CE}$ is mediated via the EA. We find no significant causal links when using the early or full periods.

Next, we test the sensitivity of the detected causal links between spring SST index and summer SLP to slight differences in
the analysed years. We assess summer SLP using both EA and SLP indices. By removing 6 randomly selected years (12% of

tested years in the late period) in each new CEN over 500 iterations, we test whether the causal links are particularly subjected to interannual variability (Fig.3c-d). We find significant variability in the strength of the links, with a concentration around 0.25 in both cases. For $\beta_{\text{SST}_{\text{ind}} \to EA}$, a minority of samples (about 8%) exhibit negative values, highlighting the overall sensitivity of the link strength between SST and the EA pattern. This sensitivity in the causal link strength due to sampling suggests that the relationship between the spring SST index and summer SLP may be influenced by an external physical mechanism. Specifically, this could involve an additional variable not included in this CEN, such as the mechanism linking tropical SSTs described in Wulff et al. (2017).

## 3.3 Does MPI-ESM reproduce the observed link?

We now test whether the causal links detected in ERA-20C during the late period can be reproduced by MR-30. As a first step, we compare the model ability to reproduce the temporal variability of the observed summer EA. We find that MPI-ESM generally captures the range of variability, although its performance in replicating the summer EA varies across different simulation sets (Fig.4a). Historical simulations show low agreement with ERA-20C (r = 0.14), whereas MR-30 initialised simulations tend to mostly encompass the observed variability (Fig.4c).

Next, we evaluate the model skill in reproducing the spring SST index - summer EA relationship. We find that both historical and MR-30 simulations show limited skill, particularly in the late period (Figs.4b, 5). A comparison between correlation maps computed for the evaluated periods shows that while historical simulations do not show agreement in the spatial pattern of the spring SST - summer EA relationship against observations (Fig.A1a-c), the MR-30 ensemble mean shows an improvement in reproducing the mechanism (Fig.5d-f). These results motivate us to assess whether the model is able to reproduce any of the observed causal links, or whether it shows different causal paths than those observed.

The observed disparities between the model and observations, as highlighted in the spatial correlations and time series analyses depicted in Figs.4-5, prompt further investigation into the causal relationships within MR-30. To address this, we proceed to assess whether the model reproduces any of the observed causal links or presents alternative causal pathways. We construct three different CEN sets to evaluate, respectively, pi-Control, historical and initialised simulations with MR-30. The variables analysed in the CEN sets are SST, EA and SLP indices and the time lag of interest is spring - summer (3 and 4 months lag). While no causal links are found in the historical simulations, we find opposite causal links than those in ERA-20C for the pi-Control simulation, suggesting an atmospheric forcing from EA into the extratropical North Atlantic (e.g. $\beta_{EA \to \text{SST}_{\text{ind}}} \approx 0.22$), but no detected causal influence from the ocean on the atmosphere (Fig.6c).

Moving on to the initialised simulations, we leverage the entire 30-member ensemble of MR-30 to construct a comprehensive CEN spanning the full period (1908-2008), resulting in each constructed time series comprising 3030 years. We find that MR-30 is able to reproduce a weakly positive SST index - EA link (i.e. $\beta_{\text{SST}_{\text{ind}} \to EA|\text{SLP}_{\text{ind}}} = 0.04$) at 3-month lag (Fig.6a), but not at 4-month lag as detected in ERA-20C during the late period, and in much weaker strength (i.e. 0.22). Moreover, we find a weak negative causal link from SST index to SLP index in the model (i.e. $\beta_{\text{SST}_{\text{ind}} \to \text{SLP}_{\text{ind}}|EA} = -0.02$), as opposed to observations (i.e. 0.21, Fig.3b). This finding aligns with Fig.5d-f, which shows that the area of positive correlations in MR-30 is displaced southwestwards with respect to ERA-20C. No causal links from SST index to EA or SLP indices are found when analysing

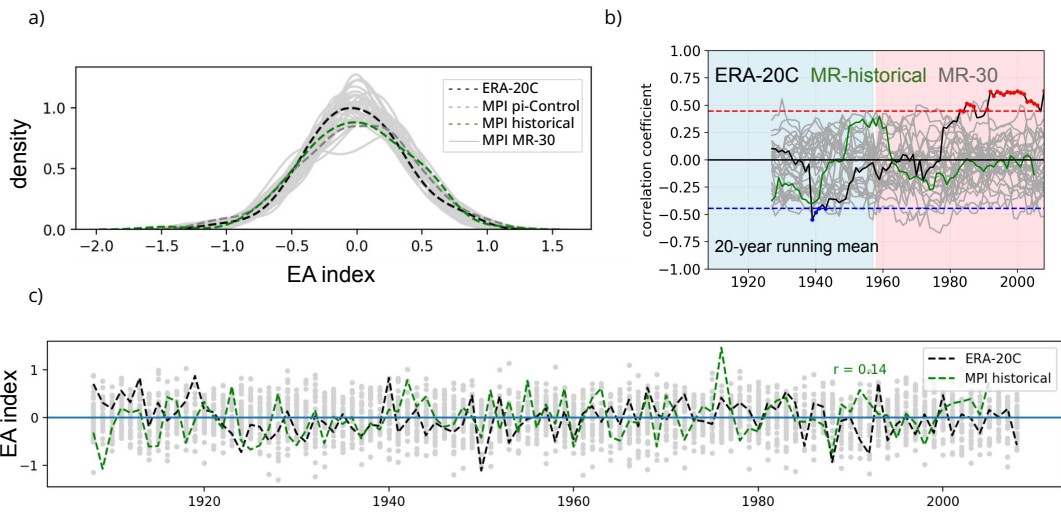

**Figure 4.** Model skill in reproducing summer EA and its link with spring SST. a) probability density functions (PDF) of the summer EA, b) running-correlation between SST and EA indices for a 20-year window, c) time series of the summer EA. Light grey colours represent individual ensemble members, black represents ERA-20C, green represents the historical simulation. In a) the dashed grey line shows the pi-Control and in b) the coloured markers indicate significant correlations at the 95% confidence interval, illustrated by the horizontal dashed lines.

only the late period (1958-2008). Next, we therefore investigate the causal link sensitivity to the sample size and focus on 45-year long time series covering the late period, allowing a direct comparison with the sensitivity analysis performed in ERA-20C (Fig.6b-d).

### 3.4 Sensitivity analysis and impact on predictive skill

We perform a two-step sampling method in our sensitivity analysis with MR-30 for SST, EA and SLP indices (Sect.2.4). Our
sensitivity results suggest that the model predominantly fails to reproduce the observed links between SST index and EA or SLP indices (Fig.6b), showing only in about 5% of the cases $\beta$-coefficients in the positive range as in ERA-20C (Fig.3).

We hypothesise that this MR-30 limitation in reproducing the causal links detected in ERA-20C might constrain the skillful prediction of European summers a season ahead. As a first test, we focus on two particular values of the $\beta$-coefficients, namely $\beta_1 = -0.18$ and $\beta_2 = 0.18$, corresponding to the link SST index $\rightarrow$ SLP at 3-month lag illustrated by orange arrows in Fig.6b. In
other words, we analyse two cases with strong causal link strength but in opposite signs, with $\beta_2$ lying closest to the observed ERA-20C range.

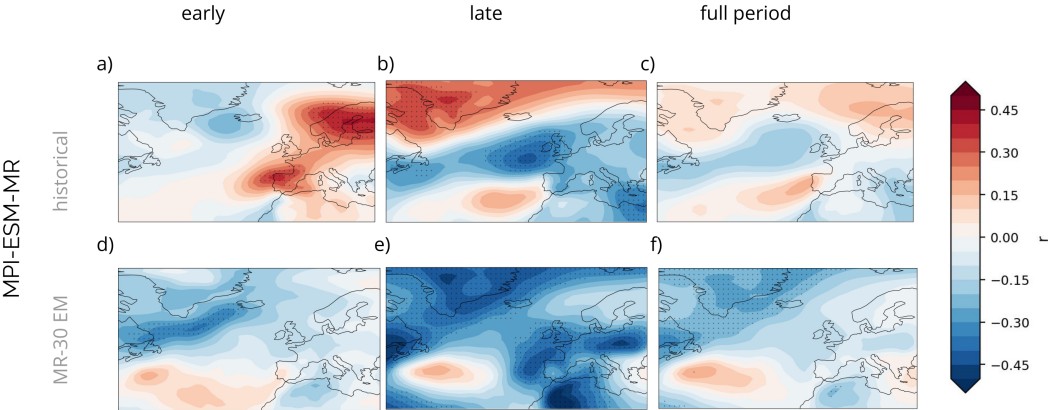

**Figure 5.** Spatial characteristics of the SST-SLP relationship over the 20th century in MPI-ESM-MR. Correlation maps show point-wise correlation coefficients for the April-May SST index and July-August SLP means considering *early* (1908-1957; a,d), *late* (1958-2008; b,e) and *full* periods (1908-2008; c,f), respectively. Top row shows results for the MPI-ESM-MR historical simulation and bottom row for MPI-ESM-MR 30-member ensemble. The reader may refer to Figs.A1a-c for a comparison with ERA-20C.

We perform a predictive skill assessment for the MR-30 bootstrap ensemble respective to $\beta_1$ and $\beta_2$ against ERA-20C, checking whether the strength of the causal link has a fingerprint in the predictive skill of JA SLP (Sect.2.5). We find a better agreement between model and reanalysis for $\beta_2$ than for $\beta_1$, with significant ACC particularly over the region where spring SST is significantly correlated to summer SLP in ERA-20C (e.g. FigA1b). However, since positive causal links are only rarely present in MR-30, we are unable to identify a robust fingerprint in the predictive skill related to any of the links between SST and EA or SLP indices.

### 3.5 Forecasts of opportunity: could causality help?

We aim to identify a robust fingerprint of spring NA-SST on summer predictive skill, which could potentially enhance targeted forecasting opportunities (Mariotti et al., 2020). Our correlation analysis, as depicted in Fig. A1, indicates the potential influence of spring NA-SST on summer T2m variability across the Euro-Atlantic region during the late period. Thus, we conduct an additional causal analysis in ERA-20C to pinpoint the regions within the T2m field where a causal relationship with spring NA-SST is anticipated. We also explore whether this causal relationship might impact the predictive skill of MR-30.

We compute a causal map (Di Capua et al., 2020b) that represents the $\beta$-coefficients calculated for the link between AM SST index and each grid point of JA T2m and SLP fields conditioned on the SLP index, i.e. $\beta_{\text{SST}_{\text{ind}} \rightarrow \text{T2m}|\text{SLP}_{\text{ind}}}$ and $\beta_{\text{SST}_{\text{ind}} \rightarrow \text{SLP}|\text{SLP}_{\text{ind}}}$ (Fig.7b, shading and contours, respectively). The choice of using either the EA or SLP index has minimal impact on the results, as shown by the similar causal map generated with the EA index in Appendix A Fig. 2. We find two causal regions of opposite signs. The first region shows negative causal links and is located in northwestern Europe, partly encompassing the area used to

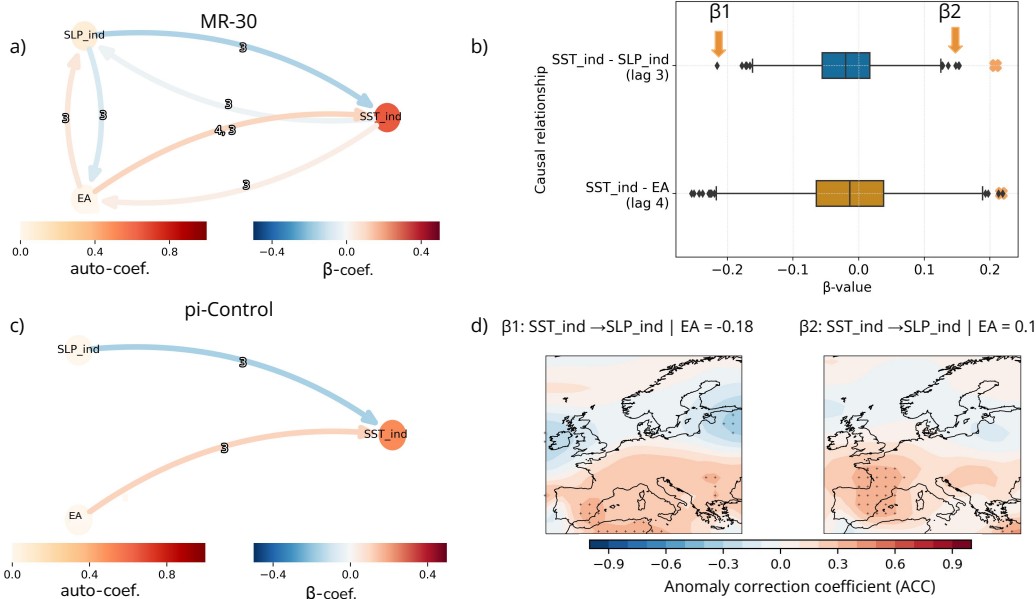

**Figure 6.** Causal Effect Networks for MPI-ESM. a) CEN between SST index, EA teleconnection and SLP index for the MPI-ESM-MR 30-member ensemble (MR-30) considering the full period. b) Sensitivity of the causal links between SST and SLP indices at 3-month lag, and SST and EA indices at 4-month lags in the late period. Boxplots show $\beta$-coefficients calculated for a random selection of 45 years, sampling one random ensemble member amongst the 30-member set per year. This process is repeated 1000 times and only significant $\beta$-coefficients are shown (p-value < 0.1). Orange "x" markers represent the $\beta$-coefficient calculated from ERA-20C (dashed lines in Fig.3). c) Same as (a) for a 1000-year long pi-Control simulation with MPI-ESM-MR. d) Comparison of the impact on SLP predictive skill in lead times of 3-4 months in MR-30 against ERA-20C for time series showing opposite $\beta$-coefficient strengths: a MR-30 bootstrap ensemble with (left) $\beta_1$ = -0.18, and (right) $\beta_2$ = 0.18. Predictive skill is quantified with anomaly correlation coefficients for the late period. $\beta_1$ and $\beta_2$ are highlighted in (b) by orange arrows.

calculate the T2m$_{CE}$ index expressed in the causal graph in Fig.3a, hereafter *CE* region. This can be interpreted as an increase

of 1 SD in the spring SST index (e.g. warming over subpolar, and cooling over subtropical North Atlantic) causally driving a decrease of about 0.3 SD in the summer T2m field in northwestern Europe. The second region shows a positive causal influence on both T2m and SLP fields, reaching strong values above 0.5 for the T2m field. A black box illustrates this causal region, denoted *Ridge* (Sect.2.2).

Targeting the two causal regions *CE* and *Ridge*, we test the hypothesis that the predictive skill of summer surface climate

in MR-30 is higher for time series that can reproduce the causal link strength observed in ERA-20C, compared to those that cannot. Thus, we specifically test the four links corresponding to the causal graphs in Fig.7a,c in MR-30. To this end, we first perform cross-validation and ensemble subsampling to generate 1000 time series for each analysed link, each timeseries consisting of 45-year periods randomly selected from the ensemble space during the late period (Fig.8a-c and SI Fig. 3). While

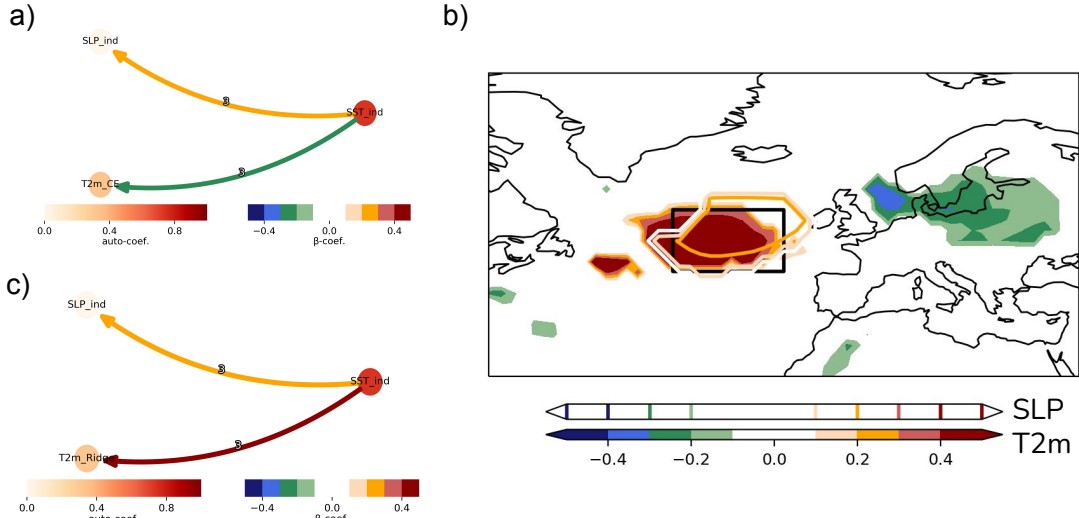

**Figure 7.** Spatial features of the causal influence of spring SST index on summer temperature. a) observed causal links between SST index, SLP index and $T2m_{CE}$ in the late period (1958-2008); b) respective causal map for 3-month lag, showing causal links between April-May SST index and July-August temperature in shading $\beta_{SST_{ind} \to T2m|SLP_{ind}}$ and April-May SST index and July-August SLP $\beta_{SST_{ind} \to SLP|SLP_{ind}}$ in contours. Black box highlights the region of strongest causal influence and represents the area used to calculate the T2m index denoted $T2m_{Ridge}$ in the text; c) observed causal links between SST index, SLP index and $T2m_{Ridge}$ in the late period (1958-2008).

MR-30 mostly fails to encompass the observed link strengths, we focus on the extremes: time series that lie near the tails of
the distribution. Specifically, we examine the percentiles closest to the observed link strength, looking at the 95th percentile for values near positive observations and the 5th percentile for negative observations. Next, we evaluate whether the strength of the causal links is imprinted on MR-30's skill in predicting summer SLP, T2m and Z500 for the two causal regions a season ahead. We quantify the predictive skill with ACC using ERA-20C as a reference. We calculate the ACC for each of the total 50 samples, averaging over the Ridge region (Fig.8d,f) and CE region (Fig.8e). In panel d, we observe that the ACC for SLP and
Z500 is significantly higher when time series are closest to the observed values (i.e., above the 95th percentile), as indicated by non-overlapping boxplot envelopes. The difference in T2m ACC is more subtle, with a slightly higher median and higher upper quartile for time series closer to the observed values. For panels e and f, differences are minimal, with only a slight increase in ACC for time series closer to the observed values. In conclusion, while MR-30 generally struggles to capture the observed link strengths, time series that align more closely with the observed values show a significant improvement in predictive skill
for SLP and Z500. This suggests that achieving closer alignment with observed causal link strengths can notably enhance predictive performance for these variables, though the effect is less pronounced for T2m than SLP or Z500 in both analysed regions.

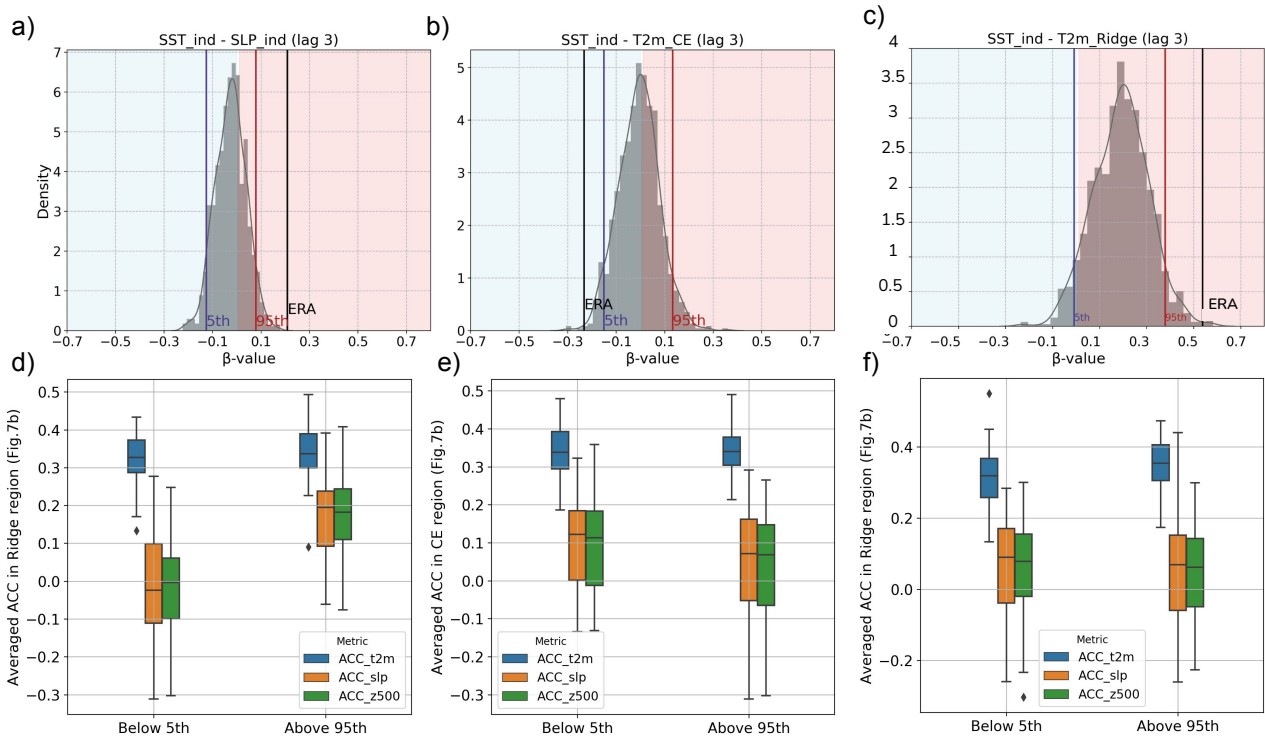

**Figure 8.** Influence of Spring (AM) SST Index on Summer (JA) Predictive Skill in MR-30. The upper row shows the sensitivity of causal link strength in MR-30 for a) $\beta_{\text{SST}_{\text{ind}} \rightarrow \text{SLP}_{\text{ind}} | \text{T2m}_{\text{CE}}}$, b) $\beta_{\text{SST}_{\text{ind}} \rightarrow \text{T2m}_{\text{CE}} | \text{SLP}_{\text{ind}}}$, and c) $\beta_{\text{SST}_{\text{ind}} \rightarrow \text{T2m}_{\text{Ridge}} | \text{SLP}_{\text{ind}}}$. Each distribution is generated from 1000 bootstrap samples, with a random selection of 45 years and one ensemble member from the 30-member set per year. The lower row compares the predictive skill of MR-30 against ERA-20C, highlighting time series that best match the observed causal link strength (above the 95th percentile for positive values, below the 5th percentile for negative values) versus those furthest from the observed strength (below the 5th percentile for positive links, above the 95th for negative links). Mean ACCs are shown for July-August sea level pressure (SLP), 2-metre air temperature (T2m), and 500 hPa geopotential height (Z500), averaged over the region highlighted by the grey box. Further details are provided in Sect.3.5.

## 4 Discussion

The framework of *forecasts of opportunity* (Mariotti et al., 2020) in seasonal prediction has been increasingly explored to identify physical processes which lead to enhanced predictability and forecast skill. Such a strategy has been particularly useful for summer (Carvalho-Oliveira et al., 2022) and winter (Dobrynin et al., 2018) seasonal predictions in the European region, where predictive skill is limited. Here, we target the summer EA to understand how its seasonal predictability is influenced by spring North Atlantic SSTs using the causal inference-based tool CEN based on PCMCI algorithm.

Using ERA-20C, our CEN analysis confirms that the spring SST index proposed in Ossó et al. (2018) causally influences the variability of summer SLP in the Euro-Atlantic region with a 3-4 months delay during the late period (1958-2008). Specifically, we find that a 1 SD change in the spring SST index first drives a 0.2 SD change in the summer SLP index at 3-month lag (e.g. March-April SST index → June-July SLP index), and then drives a 0.2 SD change a month later in the summer EA (e.g. March-April SST index → July-August EA, Fig.3a). While EA and SLP indices are highly correlated (r = 0.82), the position of the area used to calculate the SLP index (Fig.A1c) only partly overlaps the EA centre of action (Fig.1a), which extends further northwest. We speculate that the northward migration of the North Atlantic jet stream during summer (e.g. Hallam et al. (2022)) could explain the delay of a month between the causal link of SST index and EA/SLP indices.

Besides extratropical SSTs, ENSO-related tropical forcing has been suggested to influence the summer EA over more recent decades (1979 - 2016, e.g. Wulff et al. (2017); O'Reilly et al. (2018)). As opposed to the mechanism proposed in Ossó et al. (2018), Wulff et al. (2017) suggested that the summer EA is forced by diabatic heating anomalies in the tropical Pacific and Caribbean, and it is characterised by an extratropical Rossby wave train with a centre of action west of the British Isles. The CEN analysis proposed in this paper could therefore be extended to include tropical SST predictors, thus testing how the causal links discussed here could be affected by the influence of additional drivers.

Our findings suggest that the causal links detected in ERA-20C are nonstationary during the 20th century, being present only in the late period (1958-2008). Nonstationarity in teleconnections has been reported by several studies (e.g. Woollings et al. (2015); Weisheimer et al. (2019)). In particular, Rieke et al. (2021) used a 700-year pre-industrial control run with MPI-ESM-LR to investigate the tropical link of the summer EA (Wulff et al., 2017) with a statistical model, and showed that the link had a nonstationary behaviour, being present in some multidecadal epochs but not in others. Detecting nonstationarity in the causal links discussed here has an important consequence for the application on predictive skill in seasonal forecasting, implying a limited use of such causal links to target forecasts of opportunity.

Yet, our causal analysis with CEN offers an alternative assessment of MPI-ESM-MR's performance, enabling a direct comparison of the causal links reproduced by the model with those detected in reanalysis. We find that the causal links between spring SST index and summer EA and T2m are absent in pi-Control and historical simulations, but appear in some 45-year-long time series sampled in the initialised ensemble MR-30, thus suggesting a role of initialisation (Fig.6). Nevertheless, our results suggest that MR-30's limited performance in reproducing these causal links, in particular between spring SST index and the summer EA, might explain its low skill in predicting summer seasonal European climate (e.g. Neddermann et al. (2018); Carvalho-Oliveira et al. (2022)).

## 5 Conclusions

We apply the causal inference-based tool CEN based on PCMCI algorithm to evaluate the influence of spring North Atlantic extratropical SSTs on the predictability of summer EA and its associated impact on surface climate at seasonal timescales. Our main findings are:

- Analysing ERA-20C, we find that the observed relationship between spring SST index and summer EA is nonstationary during the 20th century, showing distinct spatial patterns between early (1902-1957) and late (1958-2008) periods. The estimated causal influence of spring SST index on summer EA is of $\beta \approx 0.2$.

- We find that this relationship in ERA-20C is only causal over the late period. A sensitivity analysis of its strength during the late period shows high variability, suggesting that the presence or absence of specific years plays an important role in the quantification of the causal link. This may suggest that an external physical mechanism not included in our analysis might modulate the spring SST - summer EA causal link.

- In addition to summer EA, we find in ERA-20C that the spring SST index causally influences summer T2m ($\beta \approx$ -0.2) over a region in northwestern Europe, with about a third of this causal influence being mediated by the EA.

- We find that pre-industrial and historical simulations of the MPI-ESM-MR do not reproduce the causal links detected in ERA-20C during the late period. In contrast, our CEN analysis with the full initialised ensemble MR-30 reveals a weak positive causal link between spring SST index and summer EA ($\beta \approx 0.04$).

- However, for 45-year-long time series randomly sampled in MR-30, we find that the initialised ensemble is mostly unable to reproduce the spring SST index - summer EA link. Despite this, there are notable exceptions where individual time series that lie closer to the observed causal link exhibit significantly improved predictive skill.

- This improvement is particularly evident for 3-4 month lead time SLP and Z500, where higher skill is associated with time series that capture the observed causal link strength more accurately. These results suggest that even within a generally underperforming ensemble, there are instances where a closer alignment with observed causality leads to more skilful predictions, especially for key atmospheric variables.

In this analysis, we demonstrate that MPI-ESM-MR has limited performance in reproducing a causal link between spring NA-SST (SST index) and summer EA amongst uninitialised and initialised model datasets. Our causality analysis therefore sheds light on the limitations of this model in providing skillful seasonal predictions of summer climate, particularly over areas which undergo a significant EA influence. Addressing these deficiencies, such as inadequacies in representing crucial coupled ocean-atmosphere feedbacks, will be key in future model improvements. Finally, our results for the initialised ensemble MR-30 show that ensemble members able to reproduce a causal link to spring SST have a potential for regional skill improvement. This highlights how causality frameworks can target forecast opportunities, and emphasises the importance of enhancing the representation of teleconnections in climate models.

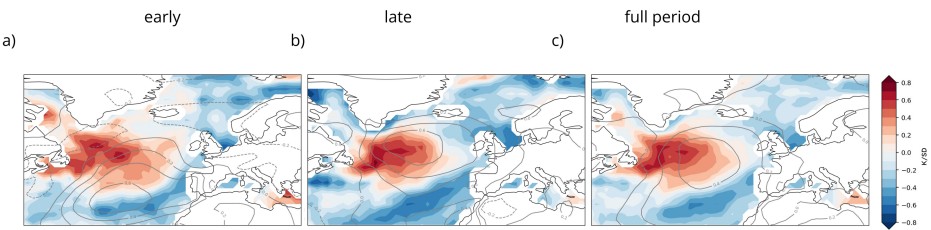

early    late    full period

**Figure A1.** Linear regression maps of July-August NA-SST anomalies (shading) and SLP (contours) against the precursor March-April SST index for ERA-20C (normalised by the SD), for a) *early* (1908-1957), b) *late* (1958-2008) and *full* periods (1908-2008). Contour interval is 0.2 hPaSD$^{-1}$.

*Data availability.* Model simulations were performed using the high- performance computer at the German Climate Computing Center (DKRZ). All data are stored at the DKRZ in archive and can be made accessible upon request (www.dkrz.de/up). The Reanalysis of the 20th Century provided by the European Centre for Medium-Range Weather Forecasts (ECMWF) was used in this study, downloaded from https://www.ecmwf.int/en/forecasts/dataset/ecmwf-reanalysis-20th-century, with access prior to its discontinuation on 1 June 2023.

## Appendix A: Appendix A

Appendix A provides supplementary figures that offer additional context and support for the main analysis presented in the manuscript.

*Author contributions.* JCO, GDC, LFB, RVD and JB contributed to the design of the analysis. JCO performed the analysis and wrote the first draft of the paper. JCO, GDC, LFB, RVD and JB contributed to the interpretation of the results and to the writing of the paper.

*Competing interests.* The contact author has declared that none of the authors has any competing interests.

*Acknowledgements.* The authors would like to thank Efi Rousi and Eduardo Zorita for insightful discussions at the start phase of this research. The authors also thank the Climate Modelling group at Universität Hamburg and the Climate Extremes group at VU Amsterdam for helpful discussions. Model simulations were performed using the high-performance computer at the German Climate Computing Center (DKRZ). This research was funded by the Deutsche Forschungsgemeinschaft (DFG, German Research Foundation) under Germany's Excellence Strategy—EXC 2037 'CLICCS—Climate, Climatic Change, and Society'—Project Number: 390683824, contribution to the Center for Earth System Research and Sustainability (CEN) of Universität Hamburg (JB and LFB), and by the German Federal Ministry for Education

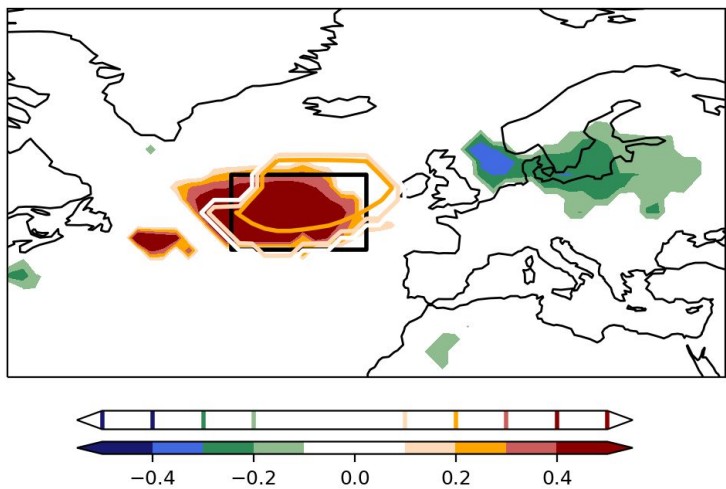

**Figure A2.** Causal map for 4-month lag, showing causal links between March-April SST index and July-August temperature in shading $\beta_{\mathrm{SST_{ind}} \to \mathrm{T2m|EA}}$ and March-April SST index and July-August SLP $\beta_{\mathrm{SST_{ind}} \to \mathrm{SLP|EA}}$ in contours. Black box highlights the region of strongest causal influence and represents the area used to calculate the T2m index denoted $\mathrm{T2m}_{Ridge}$ in the text.

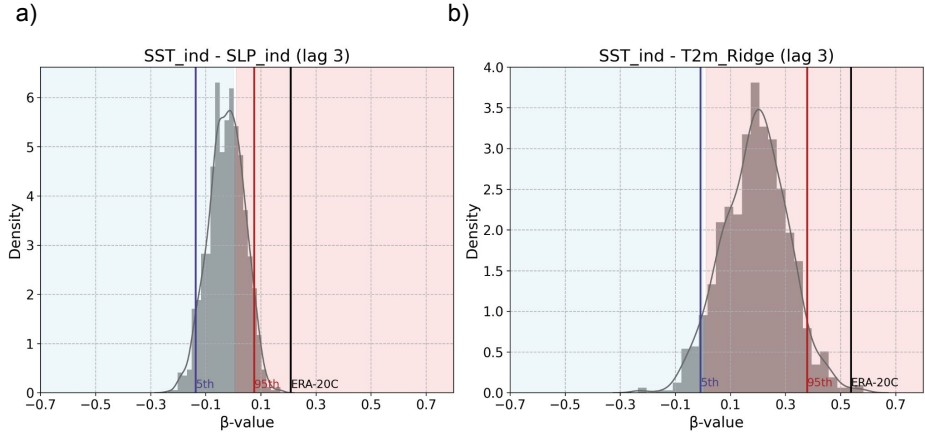

**Figure A3.** Sensitivity of causal link strength in MR-30 with respect to the causal graph in Fig.7c. a) $\beta_{\mathrm{SST_{ind}} \to \mathrm{SLP_{ind}|T2m_{Ridge}}}$, b) $\beta_{\mathrm{SST_{ind}} \to \mathrm{T2m_{Ridge}|SLP_{ind}}}$. Each distribution is generated from 1000 bootstrap samples, with a random selection of 45 years and one ensemble member from the 30-member set per year.

and Research of Germany (BMBF) via the JPI Climate/JPI Oceans project ROADMAP (grant no. 01LP2002B) (GDC and RVD). JB was
supported by Copernicus Climate Change Service, funded by the EU, under contracts C3S-330, C3S2-370.

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
