# Peer review of "Causal relationships and predictability of the summer East Atlantic teleconnection"

_EGUsphere, 2023_

## Referee Comment (RC1)

**General comments.**

The manuscript addresses the relationship and causal connection between spring North Atlantic sea surface temperature and the summer East Atlantic pattern in reanalysis and MPI-ESM-MR model simulations during the 20th century, which fits the scope of the WCD journal. This study uses correlation and linear regression maps to primarily estimate the relationships among various variables, and then uses causal discovery to detect cause-effect relationships. First, such analysis is performed for ERA-20C reanalysis, and then the results are compared to MPI-ESM-MR simulations. In my opinion, the authors address the novelty of the application of causal discovery in their research area, and clearly present their results in clear and fluent language, however, several major issues need to be addressed.

Methodology. The authors provide a brief overview of the used methods, in particular causality, via referencing Runge et al., 2015, Kretschmer et al.,2016, Di Capua et al., 2020b. The authors mention the "causal inference-based tool", but it is not clear which tool is meant. The authors do not mention particular settings used in the causal discovery algorithm, such as $\tau_{min}$, $\tau_{max}$, pc_alpha, alpha_level, conditional independence test etc. Without these parameters, it is very hard to reproduce particular results.

While, for example, in L193 the authors indicate that their analysis focuses on 3 and 4 months lag only (guessing $\tau_{min}$=3, $\tau_{max}$ =4), it is not clear from the context of the paper why contemporaneous links were not included.

The authors used different variables for reanalysis and model simulation to construct their causal graphs/CENs and then compared outcomes. I address this issue in more detail in "Specific comments", but I highly recommend that the authors stay consistent in their analysis, especially in drawing proper conclusions.

Naming of variables. The authors use SST to indicate the North Atlantic extratropical surface temperate (P1L2) or SST index (P1L4) for the meridional SST gradient over the North Atlantic region. I highly recommend that the authors use the SST term to indicate SST and include a geographical indicator to their variable, e.g. NA-SST, when they refer to the North Atlantic extratropical surface temperate.

The term "causal associations", apart from the title, is used twice in the manuscript. I suggest the terms "relationships/links/connections".

**Specific comments.**

**Abstract**.

P1L6 "We only find this link to be causal, however, during the period 1958 - 2008." → the authors did not mention the analyzed period for ERA-20C reanalysis therefore "however" in L6 sounds odd. Moreover, later in the manuscript the authors provide the reasoning and explanations why there are no detected causal links for specific periods. Therefore, I suggest to keep the sentence simple "We find the causal link during 1958-2008".

P1L9 "we find that spring SST" → do the authors mean spring North Atlantic SST? "NA-SST" would be helpful to follow the text. In P1L4 the authors introduced the SST index, so do the authors talk about SST index here as well?

P1L14 What is the meaning of "moderately" here?

P1L13 "We find that while MPI-ESM-MR…"→ both pre-industrial and historical simulations?

**Introduction**.

P2L26 What is this causal inference-based tool?

P2L40 SST is already introduced in P2L27. I refer the authors to my comment to improve the notation of the North Atlantic extratropical surface temperate instead of simply using SST.

P3L55 Runge et al., 2015 does not use term "Causal Effect Networks". It was Kretchmer et al., who was one of the first authors to use this term.

P3L58 regarding overcoming spurious correlations: see also Runge et al., 2014; Runge, Bathiany, et al., 2019.

P3L59 please add more examples for the application of causal discovery for other teleconnections. For example, Atlantic-Pacific teleconnections: Karmouche et al., 2023; Arctic-midlatitude teleconnections: Kretschmer et al., 2020, Siew et al., 2020; Galytska et al., 2023, marine cold-air outbreaks: Polkova et al., 2021, Walker circulation: Runge, Bathiany, et al., 2019 and others.

**Methodology**.

P3L73 Here SST stands just for Sea Surface Temperature. That is confusing with the North Atlantic SST introduced before.

L106 EOF is already introduced. The calculation of EA index is already mentioned in the Introduction.

L114. The reference to Fig. 1b comes before Fig. 1a. What is the meaning of the colors in Fig. 1b? Please fix this. Generally, I suggest that the authors insert the Figure and/or Table, which would summarize/show the indices used in their studies and explain them in Sect. 2.2. In that case panels c-f from Fig. 1 would solely address the discussions from Sect. 3.1

L117 The authors explain how they calculated an SLP index, however the introductions lacks the explanation of the impact and significance of the pressure over this region on East Atlantic Pattern and North Atlantic SST.

L121 "wherever useful"→ where exactly?

L127 Cite Spirtes et al., 2000 for PC part of the algorithm.

L134 "circles" → nodes?

L137 The authors should also explain the meaning of the color of the nodes.

L141-142 I assume the authors used Tigramite for their research. I find it important that the authors stay transparent on the software that they used as well as the settings that were applied, $\tau_{max}$, $\tau_{min}$, pc_alpha, alpha_level. I am wondering if the authors already tested PCMCI+ algorithm for their study, which is able to capture causal contemporaneous connections. Why authors did not include contemporaneous connections?

**Results**

P6L146-147 If the authors follow my suggestion and introduce a separate Figure for Sect. 2.2 and summarize the used indices, I suggest to move this sentence to Sect. 2.2.

P6L147-149 Fig. 1 c and d do not show "below average temperature/precipitation", they show correlation between EA-index and temperature/precipitation, which is associated with below-average temperatures/precipitation over this region.

P6L149-150 This sentence should be moved to Sect. 2.2. The simple usage of term "SST index" is very confusing.

P6L153 "correlation reaches significant values" → what is the definition of significance here? Did the authors use significance test here?

Figure 2. Caption. I suggest: "Distinct spatial characteristics of the spring SST influence on the summer circulation over the 20th century (for ERA-20C) for early (1908-1957, left), late (1958-2008, middle) and full periods (1908-2008, right column). …" …"Box in Fig.2i illustrates…". To label y-axes the authors use "Influence of AM SST index", which implies the direction of the impact, however by this point it was not yet discussed. I suggest terms "relationship/connection" instead. Another minor suggestion: if the authors draw coastlines in gray, then SLP contours in panels d-f could be plotted in black and become more visible. Further, since Fig. 5 shows only correlations (models vs reanalysis), I suggest that the authors similarly restructure the panels in Fig. 2, e.g. first two rows showing correlations, the third raw showing regression.

L166 Is it about panels a and b?

L166-167 Not shown in this paper? Or which Figure/panels?

L171 But not in panel d?

L173-174 In which geographical region?

P9L191 "correlation" → anticorrelation

L193 I assume $\tau_{min} = 3$, $\tau_{max} = 4$.

L200-201. In regard to the full periods, the authors already stated that the spring SST -summer EA relationship is nonstationary for the full period, which would lead to the non-realistic causal graph/CEN (also see Runge 2018).

L208 "actor" → variable. The authors did not define the term "actor", but used term "variable" throughout the manuscript.

L208 excluded → "not included".

Figure 3. Given that panel a consists of two subfigures that aim to show the same result, I strongly recommend that the authors combine them into one figure. For example, in plot_graph function, the authors can use node_pos.

Figure 5. Since panels g-h are the same as Fig. 2a-c, the authors can either remove these panels completely, or show the differences instead.

P9 L222 I do not recommend to comparing causal graphs/CEN between reanalysis and model simulations while using different sets of variables. The assumption of causal sufficiency says that *Measured variables include all of the common causes.* For the reanalysis, the authors motivated the usage of 2m Temperature. It is important to explain, why authors did not include this variable for model simulations. For example, the correlation of AM SST index and 2m Temperature is not that strong in the model. But this might lead to the other discussion, e.g, low correlation could be a consequence of different state(s) of AM SST index in the model vs reanalysis. Currently the results from Fig3a and Fig.6a,c are not directly comparable.

P9 L222-223 Here the authors indicate that for the model simulations the time lag was 3 and 4 months, however Fig. 6a also shows two causal lagged links with 2 months delay. Please, clarify this.

P10 L224 "suggesting an atmospheric forcing into ocean" → please, also specify exact links. This will improve readability and comparison with the results from Fig. 6. Same for L225.

P10 L227-230 I would like to highlight that differences between reanalysis and model simulations were expected, since the authors already showed that the timeseries (Fig.4) as well as spatial patterns differ quite a lot between different data sources. I would appreciate if the authors addressed this issue here (as well as in the Discussion).

L232 There is no Fig. 6e

Figure 6 a and c. Node "EAP"→ EA.

Figure 7a. It is not clear why the authors did not keep SLP-ind in the causal graph? Is there a reason why the colormap for the β -coef. is changed?

**Discussion**

P15LL288 For which time period?

**Conclusions**

P16 L323 "we find that"… but in L303 the authors state that the nonstationarity has been previously reported by other studies.

L326 Please, specify if this conclusion is for the reanalysis. Same for L330.

L328 "that an external physical mechanism not included in our analysis" → the usage of PCMCI implies that the user makes an assumption that all variables are included to represent a specific mechanism. If user does assume that all variables are included in the causal graph/CEN, then LPCMCI should be rather used.

L333. Please, also summarize why the model does not reproduce the links from the reanalysis.

L334 "weakly"→ weak

L341 "limited performance in reproducing"→ I suggest to highlight the reasons while giving such a statement.

---

## Referee Comment (RC2)

Review of "Causal associations and predictability of the

summer East Atlantic teleconnection"

by Carvalho-Oliveira et al.

The paper adopts the quite novel technique of Causal Effect Networks (CEN) to analyze the link between spring SSTs in the North Atlantic (northwest-southeast difference) with the East Atlantic (EA) pattern in SLP in summer. This is done both for the ERA20C reanalysis and for the MPI model, in various configurations. Moreover, it is analyzed whether the representation of such causal link influences the model forecast skill on a specific region.

The work is well framed in the introduction and the proposed CEN also quite well justified in terms of possible physical pathways in the introduction, though not much in the rest of the paper. The methodology is quite complex and involves a large amount of work for a challenging analysis. The results are interesting, but I think they could be presented in a clearer way, at least in some parts. Also, the nomenclature is quite complex to follow due to the large number of variables, experiments, techniques considered: some more effort on this issue would help the reader better grasp the main results of the work.

Main issues, mostly related with the presentation of the results:

- Period. MA, AM, MAM? It is quite difficult to follow the various spring selections and to understand why one is chosen with respect to the other. As long as I understood, most results refer to AM, but the other seasons are cited here and there. I think this is quite confounding for the reader, and a single spring window would help, maybe just stating that changing it does not change the results.

- The methods section 2.2 could be expanded to include a more detailed description of the bootstrap ensemble, which is currently just in the main text. It is unclear how the authors refer to the bootstrap ensemble throughout the text. Linked to this, the term "causal ensemble" or "causal timeseries" is not explained in the text.

  Also, since many indexes are considered in the text, a point-by-point list of all indexes considered with a unique and identifying name would help. For example the SST_Ridge just appears at the end of the results section, but would be useful to have it here for quick reference.

- The end of the results section could possibly make for a separate subsection: see comment at lines 256-281. I think this is one of the most interesting results of the paper, but is currently difficult to grasp and could be easily lost in the main text.

- Physical pathway. The CEN framework is a powerful tool but should be used with caution. In particular two things are necessary, following Kretschmer et al. (2021):

  - the CEN should only be used to "measure" a causal link for which there is already an hypothesized physical pathway;

- the causality of the link is always conditional to the choice of the variables included in the model, meaning that if a relevant variable is missing, the CEN result may be wrong.

I think the authors respect both "requirements", but I suggest to:

- recall the possible physical mechanism behind the link when presenting the CEN. It is currently only cited in the introduction.

- put more emphasis and discuss on the possible impact of a missing process in the CEN.

Specific points.

L94. The ocean state is derived  *from?*

L94-97. It is not completely clear to me how the assimilation experiment is performed for the ocean. If is said at line 94 that an ocean-only simulation with MPI-OM forced with ERA-20C is performed. Is this ocean state then used for the nudging of the 30 ensemble members? I think this choice should be (quickly) motivated in the text.

L102. ..at lead times *of* 3-4 months..

L106. I would say: "... the second principal component (PC) of the  empirical orthogonal function (EOF) *decomposition* of ..."

L131. Since the technique is pretty novel and can easily generate misunderstandings, I suggest the authors to add some further disclaimer for the reader. In particular, as implicitly stated a few lines above, a limitation of the Causal effect network analysis is that the choice of the variables to be considered is crucial for determining the causality of the link. In this sense, the possibility of a spurious correlation can never be completely excluded. I think this should be made clear in the text, expanding the sentence at L131.

Figure 1 caption. Specify the period considered for the SST anomaly. From the text it is spring SSTs and summer EA (L151), but this is not clear for the caption. Also, it is not clear what period panels c, d, e are referring to. In general, since different periods are considered for different variables, a suggestion would be to put the period as a subscript: $SST_{MA}$.

L151. Linked to the above. Is the spring SST referred to MAM, MA or AM?

L166. Why not use MAM? Is there some process changing significantly between early and late spring?

L200. The T2m$_{CE}$ looks like potentially correlated with EA with no lag (from the composite in Fig. 1).. isn't this correlation appearing in the CEN?

L207-8. I appreciate that the author acknowledge that something could be missing from the CEN. Also, could this mean that the observed causal link may be to some extent spurious, since some key process is missing from the CEN?

L212. I wouldn't call this "skill in reproducing the summer EA" since Fig. 4a is evaluating the pdf of the EA index, so just from a statistical/climatological point of view.

L216. In what sense is Fig. 4b showing "MR-30 capturing the temporal variability of the relationship in the early period" ? I think the only information is about the spread of the relationship, but I do not see a tendency for a negative correlation in the early period as observed for ERA20C.

L221. I agree MR-30 looks slightly better for the early period, but still is quite far from ERA20C (the positive correlation in the southern North Atlantic is not significant and does not extend so much North). This is true for the ensemble mean, but have you checked whether some individual member is getting a response closer to the observed relation? This could possibly inform on the "missing" process in the chain.

L222. I do not understand "first" in the sentence.

L228. 0.03 seems very small. Is it significant?

L245. What do you mean by "MR-30 causal timeseries"? If referred to the "MR-30 causal ensemble" in Fig. 6 caption, I would change the wording to something different.. e.g. MR-30 bootstrap ensemble

L245. How do you perform the "predictive skill assessment"? By selecting random members with beta close to beta_1 and checking the skill only for those?

L249. How rare is this? The information might be relevant.

L257. I can't easily see the contours in Fig. 7b. Would be better a separate figure, or black contours.

L256-L281. I would separate this last part, since it is focussing on a different topic: how does the existence of a causal link between SST spring and T2m Ridge (a different predictand from the rest of the paper) influence the forecast skill on the Ridge region in the random MR-30 ensemble?

Also, I had a hard time following the section, which I think should be separated from the rest, better framed and possibly expanded. I say this because the result looks interesting but is quite difficult to grasp from the current text. Also, the choice of the new predictand might look like a cherry pick, but I think it could be better motivated with the fact that it is the only causal link reproduced in the random MR-30 ensemble. The question "what happens to the skill when a causal link is reproduced?" seems relevant.

L312-316. This part looks a bit technical for the discussion, I suggest to remove it.

---

## Author Comment (AC1)

**Referee 1**

We thank the anonymous referee for their valuable input and for suggesting ways to improve the clarity of the manuscript. In response to their comments, we thoroughly revised the text to reflect more nuance on the results obtained. We also added analysis and refined some of the existing findings. We now find the manuscript to be substantially improved and thank the reviewer for helping us to get there. Major points addressed were the following:
- We rewrote major parts of the manuscript to improve readability and to clarify several points raised by the reviewer, in particular in section 3.3
- We substituted panel a) in Figs. 3, 6 and 7

We provide below a point-by-point response to each comment. Please note that the referee's comments are highlighted in **bold** font, while our answers are in regular font.

**Major comments:**

1) **Methodology. The authors provide a brief overview of the used methods, in particular causality, via referencing Runge et al., 2015, Kretschmer et al.,2016, Di Capua et al., 2020b. The authors mention the "causal inference-based tool", but it is not clear which tool is meant. The authors do not mention particular settings used in the causal discovery algorithm, such as τmin, τmax, pc_alpha, alpha_level, conditional independence test etc. Without these parameters, it is very hard to reproduce particular results.**

   We specified in the manuscript that we use causal effect network based on PCMCI algorithm as our causal inference-based tool (L28-30). We also included the information on which CEN parameters have been used to perform the analysis in the Methodology section (L166-167).

2) **While, for example, in L193 the authors indicate that their analysis focuses on 3 and 4 months lag only (guessing τmin=3, τmax =4), it is not clear from the context of the paper why contemporaneous links were not included.**

   We thank the reviewer for this remark. We rewrote this sentence for clarity as "We test whether the spring SST index causally drives changes in summer air temperature over central Europe and under which circumstances this holds true. Therefore, our CEN analysis focuses on  τmin = 3 months and τmax = 4 months, which for simplicity we refer to as 3 and 4-month lags." (L236-239)

   We'd like to emphasise that our primary goal is to identify predictive power in forecasting, particularly focusing on discerning causal predictors in the North Atlantic, such as persistent SST features. That is, our focus is to identify causal drivers for the summer EA and air temperature which can provide information before or at

the initialisation of predictions typically in May. For this reason we are interested in quantifying links at 3-4 month lags, with contemporaneous links falling outside the scope of our study. Please see the reply to the specific comment L141-142 for more details.

3) **The authors used different variables for reanalysis and model simulation to construct their causal graphs/CENs and then compared outcomes. I address this issue in more detail in "Specific comments", but I highly recommend that the authors stay consistent in their analysis, especially in drawing proper conclusions.**

We thank the reviewer for raising this important point. As explained in the answers to the specific comments in lines 222, 223 and Figs. 6 and 7 (see below), we made sure that the computed causal graphs consisted of the same variables and hyperparameters, when comparing model and reanalysis.

4) **Naming of variables. The authors use SST to indicate the North Atlantic extratropical surface temperature (P1L2) or SST index (P1L4) for the meridional SST gradient over the North Atlantic region. I highly recommend that the authors use the SST term to indicate SST and include a geographical indicator to their variable, e.g. NA-SST, when they refer to the North Atlantic extratropical surface temperature.**

Thanks for this, indeed it makes things clearer, to include the SST region. We went through the text and edited accordingly, making sure that the reader better understands when we are referring to SST index or NA-SST in general.

5) **The term "causal associations", apart from the title, is used twice in the manuscript. I suggest the terms "relationships/links/connections".**

We thank the reviewer for the suggestion, we changed the term to "relationships".

**Minor comments:**

**Abstract.**
**P1L6 "We only find this link to be causal, however, during the period 1958 - 2008." → the authors did not mention the analyzed period for ERA-20C reanalysis therefore "however" in L6 sounds odd. Moreover, later in the manuscript the authors provide the reasoning and explanations why there are no detected causal links for specific periods. Therefore, I suggest to keep the sentence simple "We find the causal link during 1958-2008".**

Thanks for the suggestion. We agree that the "however" sounds odd, so we rewrote the sentences to emphasise which period is investigated, and in which part of it we find the link to be causal. "We apply Causal Effect Networks to evaluate the influence of spring North Atlantic extratropical surface temperatures (SST) on the summer East Atlantic Pattern (EA) seasonal predictability during the period of 1908-2008." (...) "Notably, this causal link is only evident during the period from 1958 to 2008, and is not observed throughout the entirety of the investigated period."

**P1L9 "we find that spring SST" → do the authors mean spring North Atlantic SST? "NA-SST" would be helpful to follow the text. In P1L4 the authors introduced the SST index, so do the authors talk about SST index here as well?**

To make it clearer, we edited L4 as "We find in the ERA-20C reanalysis that a meridional NA-SST gradient in spring (SST index) causally influences the summer EA (...)" and L9 as "In addition to the summer EA, we find that the spring SST index has an estimated causal effect of about (...)". Besides, we edited L1 as "We apply Causal Effect Networks to evaluate the influence of spring North Atlantic extratropical surface temperatures (NA-SST)".

**P1L14 What is the meaning of "moderately" here?**

Our intention was to express that the range of causal link strength values (beta-values) reproduced by resampling the 30-member ensemble encompassed the observed causal link value (Fig. 7c). We removed the word "moderately" from the sentence to not cause misunderstanding.

**P1L13 "We find that while MPI-ESM-MR…"→ both pre-industrial and historical simulations?**

Yes, so we rewrote the sentence to make it clearer as "We find that while both the pre-industrial and historical simulations using MPI-ESM-MR are mostly unable to reproduce the causal link between spring SST and the summer EA among the different datasets, the 30-member initialised ensemble can reproduce a causal link between spring SST and summer 2-metre air temperatures over a region west of the British Isles." (L13-16)

**Introduction.**
**P2L26 What is this causal inference-based tool?**

We rewrote the sentence to include which tool we used: "Here, we apply the Peter and Clark momentary conditional independence (PCMCI) causal discovery algorithm to evaluate the

influence of North Atlantic extratropical surface temperatures (SST) on the predictability of EA at seasonal timescales." (L28-30)

**P2L40 SST is already introduced in P2L27. I refer the authors to my comment to improve the notation of the North Atlantic extratropical surface temperate instead of simply using SST.**

We took care of including the reviewer's suggestion in L27 and introduced North Atlantic sea surface temperatures as NA-SST. So we rewrote this sentence as "While there is no consensus on the physical processes driving the EA, spring NA-SST (...)".

**P3L55 Runge et al., 2015 does not use term "Causal Effect Networks". It was Kretchmer et al., who was one of the first authors to use this term.**

Indeed, thanks for bringing this up. We removed Runge et al. 2015 accordingly.

**P3L58 regarding overcoming spurious correlations: see also Runge et al., 2014; Runge, Bathiany, et al., 2019.**

Thanks for the suggestion, we included both papers in the citation.

**P3L59 please add more examples for the application of causal discovery for other teleconnections. For example, Atlantic-Pacific teleconnections: Karmouche et al., 2023; Arctic-midlatitude teleconnections: Kretschmer et al., 2020, Siew et al., 2020; Galytska et al., 2023, marine cold-air outbreaks: Polkova et al., 2021, Walker circulation: Runge, Bathiany, et al., 2019 and others.**

Thanks for the recommendation. We rewrote the sentence to accommodate some of the studies as "CEN overcomes spurious correlations due to autocorrelation, indirect effects, or common drivers (Runge et al., 2014, 2019). It has been successfully used to complement hypothesis testing for other tropical and mid-latitude teleconnections in the Atlantic-Pacific region (e.g. Karmouche et al. (2023) and Indian Ocean (e.g. Di Capua et al. (2020a)), as well as in the Arctic region (e.g. Siew et al. (2020); Kretschmer et al. (2020))." (L60-64)

**Methodology.**
**P3L73 Here SST stands just for Sea Surface Temperature. That is confusing with the North Atlantic SST introduced before.**

Indeed, so we rewrote the sentence as "The physical variables analysed are NA-SST, SLP and air temperature at 2 metre height (T2m)." (L78-79)

**L106 EOF is already introduced. The calculation of EA index is already mentioned in the Introduction.**

We rewrote the sentence as "As a first step, we define a reference EA index as the second principal component (PC) of the EOF of JA anomalies of SLP over the Euro-Atlantic sector (...)". (L111-113)

**L114. The reference to Fig. 1b comes before Fig. 1a. What is the meaning of the colors in Fig. 1b? Please fix this. Generally, I suggest that the authors insert the Figure and/or Table, which would summarize/show the indices used in their studies and explain them in Sect. 2.2. In that case panels cf from Fig. 1 would solely address the discussions from Sect. 3.1**

These were good suggestions, thanks. Below we answer to each specific point separately:
- The first reference to Fig.1a now appears in L116, whereas the first reference to Fig. 1b appears in L119. We thus keep the panels in Fig.1 in the same order.
- We chose colours to illustrate the boxes in Fig.1b to facilitate the visualisation for the reader. In the text (L114-L118) we refer to these colours when explaining which regions were taken into account to calculate the indices. To make it clearer, we now include labels in the boxes;
- As suggested, we provided a table (Table 1) summarising the indices used in the paper.

**L117 The authors explain how they calculated an SLP index, however the introductions lacks the explanation of the impact and significance of the pressure over this region on East Atlantic Pattern and North Atlantic SST.**

We thank the reviewer for the suggestion, we rewrote the sentence as "To comprehensively investigate the influence of spring NA-SST on summer SLP variability, we incorporate the SLP index introduced by Ossó et al. (2018) alongside the EA index. This approach aims to address the broader significance of pressure dynamics in the region, particularly in relation to the physical mechanism proposed by Ossó et al. (2020)." (L121-123)

**L121 "wherever useful"→ where exactly?**

We rewrote the sentence as "We use a two-tailed Student's t-test to calculate the statistical significance of point-wise correlations maps." (L131-132)

**L127 Cite Spirtes et al., 2000 for PC part of the algorithm.**

Done.

**L134 "circles" → nodes?**

Done.

**L137 The authors should also explain the meaning of the color of the nodes.**

We rewrote the sentence as "We visualise the output of PCMCI in a CEN, i.e. a causal graph where nodes represent the investigated variables, arrows indicate the direction of the causal links, and colours denote the strength of these links." (L155-156)

**L141-142 I assume the authors used Tigramite for their research. I find it important that the authors stay transparent on the software that they used as well as the settings that were applied, τmax, τmin, pc_alpha, alpha_level. I am wondering if the authors already tested PCMCI+ algorithm for their study, which is able to capture causal contemporaneous connections. Why authors did not include contemporaneous connections?**

We thank the reviewer for this important question. Our primary aim in this study was to identify precursor signals in spring, specifically before the initialisation of prediction systems typically in May, that could forecast the summer EA. Thus, we concentrated on uncovering interseasonal causal links, i.e. those between spring and summer. This approach led us to exclude contemporaneous links from our analysis, as our focus was on understanding how extratropical North Atlantic SSTs in spring influence the EA teleconnection over a span of three to four months. For this reason, performing the analysis with PCMCI+, despite potential interest from a methodological point of view, would not add any physical insight to the analysed mechanism.

We included more information on the used settings in Sect. 2.3. We added "CEN analysis is a causal discovery tool which implements the so-called Peter and Clark momentary conditional independence algorithm (PCMCI, Runge et al. 2019, Spirtes et al. 2000). We specifically use the PCMCI version 4.2 from the Python package Tigramite (https://github.com/jakobrunge/tigramite)." (L136-138)

At the end of Sect. 2.3, we added "Lastly, the PCMCI parameters are chosen as follows: pc alpha = 0.2, Alpha level to print results = 0.1, τmin = 3 months and 155 τmax = 4 months, independence test = parcorr, significance='analytic', masking type 'y'." (L165-167)

**Results**
**P6L146-147 If the authors follow my suggestion and introduce a separate Figure for Sect. 2.2 and summarize the used indices, I suggest to move this sentence to Sect. 2.2.**

We included a table (Table 1) containing an overview of the indices used in the analysis, including abbreviations, long names and the geographical regions used for the calculations.

**P6L147-149 Fig. 1 c and d do not show "below average temperature/precipitation", they show correlation between EA-index and temperature/precipitation, which is associated with below average temperatures/precipitation over this region.**

We rewrote the sentence as "A typical surface climate imprint of the summer EA in positive phase correlates with below-average temperatures in continental Europe (Fig.1c) and below-average precipitation in the British Isles and northwestern Europe (Fig.1d)." (L192-194)

**P6L149-150 This sentence should be moved to Sect. 2.2. The simple usage of term "SST index" is very confusing.**

We thank the reviewer for bringing this up. Indeed, we already have a similar sentence in Sect. 2.2 (L112). Our aim here was to recap the idea, but we agree it can be confusing. Therefore, we decided to exclude the sentence.

**P6L153 "correlation reaches significant values" → what is the definition of significance here? Did the authors use significance test here?**

We thank the reviewer for this question. Yes, we calculated the significance of the correlation using a two-tailed Student's t-test. We included the p-values in the manuscript. In addition to that, upon rechecking results we came across an error. The correct correlation coefficient for the period between 1958 and 2008 is 0.43, and not 0.51. The sentence was edited as "A Pearson correlation analysis reveals a time-dependent relationship between the SST index in spring and the EA in summer (Fig. 1e). Over a span of 101 years (1908-2008), this relationship appears weak ($r = 0.22$, $p < 0.05$). However, examining the most recent 51 years (1958-2008) shows a doubling of correlation values ($r = 0.43$, $p < 0.05$). Furthermore, focusing on the latest 30 years (the period analysed in Ossó et al. 2018) results in correlation values increasing even further to 0.60 ($p < 0.05$) ." (L195-199)

**Figure 2. Caption. I suggest: "Distinct spatial characteristics of the spring SST influence on the summer circulation over the 20th century (for ERA-20C) for early (1908-1957, left), late (1958-2008, middle) and full periods (1908-2008, right column). ..." ..."Box in Fig.2i illustrates...". To label y-axes the authors use "Influence of AM SST index", which implies the direction of the impact, however by this point it was not yet discussed. I suggest terms "relationship/connection" instead. Another minor suggestion: if the authors draw coastlines in gray, then SLP contours in panels d-f could be plotted in black and become more visible. Further, since Fig. 5 shows only correlations (models vs reanalysis), I suggest**

**that the authors similarly restructure the panels in Fig. 2, e.g. first two rows showing correlations, the third raw showing regression.**

We thank the reviewer for the helpful suggestion, we rewrote the label and edited the figure panel as the reviewer suggested. Please see revised Fig. 5 in the main text.

**L166 Is it about panels a and b?**

We rewrote the sentence as "Surrounding this high correlation region, the sign of correlations is opposite between early (Fig2a) and late (Fig2b) periods." (L210-211)

**L166-167 Not shown in this paper? Or which Figure/panels?**

Not shown, so we rewrote the sentence as "We find similar results using March-April (MA) NA-SST means, only in weaker strength (not shown)."

**L171 But not in panel d?**

Exactly, not in "panel d" of the previous version (now panel g), i.e. not in the early period. To increase clarity we rewrote the sentence as "In the late period (Fig.2h), these anticyclonic conditions coincide specifically with the position of the EA centre of action, whereas this association is absent in the early period. (L216-217)

**L173-174 In which geographical region?**
We rewrote the sentence as "We find significant correlations between the AM SST index and JA T2m, showing a similar pattern of significant positive correlations west of the British Isles, as in Fig.1c corresponding to JA EA - T2m." (L218-220)

**P9L191 "correlation" → anticorrelation**

Modified.

**L193 I assume тmin = 3, тmax = 4.**

We rewrote the sentence as "Therefore, our CEN analysis focuses on $\tau_{min}$ = 3 months and $\tau_{max}$ = 4 months, which for simplicity we refer to as 3 and 4-month lags."

**L200-201. In regard to the full periods, the authors already stated that the spring SST -summer EA relationship is nonstationary for the full period, which would lead to the non-realistic causal graph/CEN (also see Runge 2018).**

Indeed, using the full period with Tigramite could lead to a non-realistic causal graph due to nonstationarity. To emphasise this point, we revised the sentence to: "While PCMCI cannot handle the nonstationarity identified in the full period (e.g. Runge et al. 2018), we find no significant causal links in the early period." This adjustment highlights our awareness of the nonstationarity issue and underscores the need for creating sample subsets (the early and late periods) to overcome the problem.

**L208 "actor" → variable. The authors did not define the term "actor", but used term "variable" throughout the manuscript.**

For consistency, we rewrote the sentence with "variable", as suggested.

**L208 excluded → "not included".**

Modified as suggested.

**Figure 3. Given that panel a consists of two subfigures that aim to show the same result, I strongly recommend that the authors combine them into one figure. For example, in plot_graph function, the authors can use node_pos.**

Upon reconsideration, we decided to modify the panel and present the causal graph without the illustration. Considering the reviewer`s point in L222, we included instead a causal graph containing the same set of variables analysed in Fig.6a,c.

**Figure 5. Since panels g-h are the same as Fig. 2a-c, the authors can either remove these panels completely, or show the differences instead.**

Following the reviewer's suggestion, we removed panels g-h from Fig.5, since these are the same as in Fig. 2a-c. We modified the caption to accommodate this change as "Spatial characteristics of the SST-SLP relationship over the 20th century in MPI-ESM-MR. Correlation maps show point-wise correlation coefficients for the April-May SST index and July-August SLP means considering early (1908-1957; a,d,g), late (1958-2008; b,e,h) and full periods (1908-2008; c,f,i), respectively. Top row shows results for the MPI-ESM-MR historical simulation and bottom row for MPI-ESM-MR 30-member ensemble. The reader may refer to Figs.2a-c for a comparison with ERA-20C."

**P9 L222 I do not recommend to comparing causal graphs/CEN between reanalysis and model simulations while using different sets of variables. The assumption of causal sufficiency says that Measured variables include all of the common causes. For the reanalysis, the authors motivated the usage of 2m Temperature. It is important to explain, why authors did not include this variable for model simulations. For example, the**

**correlation of AM SST index and 2m Temperature is not that strong in the model. But this might lead to the other discussion, e.g, low correlation could be a consequence of different state(s) of AM SST index in the model vs reanalysis. Currently the results from Fig3a and Fig.6a,c are not directly comparable.**

We thank the reviewer for raising this very important point. To overcome this issue, we included in Fig.3b a causal graph for ERA-20C using the same set of variables analysed for MPI-ESM-MR (i.e. not including 2m temperature). By doing so, the reader is able to directly compare the causal graphs between model and observations.

**P9 L222-223 Here the authors indicate that for the model simulations the time lag was 3 and 4 months, however Fig. 6a also shows two causal lagged links with 2 months delay. Please, clarify this.**

We kindly thank the reviewer for pointing this out. We unfortunately had used the wrong figure by mistake, since over the course of our analysis we also tested the effect for tau_min = 2. The correct causal graph containing the same set of variables but focusing on tau_min = 3 and tau_max = 4 is now in Fig. 6a. Below you can find a comparison between the causal graphs. The principal link (SST -> EA, lag=3) has a marginal difference between graphs, with beta = 0.04, as opposed to beta = 0.03 in the first version of the manuscript. Besides that, other qualitative differences are:
- As expected, the autocorrelation of SST_index decreases when tau_min=3, indicating that SST persistence is higher when tau_min=2
- A weak (beta=-0.02) negative link between SST_index -> SLP_index (tau=3) appears
- The EA -> SST_index link is found for lags 4 and 3.

[Figure]

**Fig. 1** *left*: Fig.6a in the first version of the manuscript, showing tau_min=2; *right*: Fig.6a in the revised manuscript, for tau_min = 3.

To convey these changes we rewrote the text (please see below in the reply to the specific comment concerning L227-230).

**P10 L224 "suggesting an atmospheric forcing into ocean" → please, also specify exact links. This will improve readability and comparison with the results from Fig. 6. Same for L225.**

To improve readability, we rewrote the sentence as "While no causal links are found in the historical simulations, we find opposite causal links than those in ERA-20C for the pi-Control simulation, suggesting an atmospheric forcing from EA into the extratropical North Atlantic (e.g. $\beta$EA→SST ≈ 0.22), but no detected causal influence from the ocean on the atmosphere (Fig.6c)."

**P10 L227-230 I would like to highlight that differences between reanalysis and model simulations were expected, since the authors already showed that the timeseries (Fig.4) as well as spatial patterns differ quite a lot between different data sources. I would appreciate if the authors addressed this issue here (as well as in the Discussion).**

We thank the reviewer for this comment. As rightly pointed out, we anticipated such disparities given the distinct characteristics of the datasets involved. Our intention while showing these results is twofold. First, causal discovery adds the information of whether other causal paths amongst SST, EA and SLP indices appear in the different model sets for time lags of 3 to 4 months. Second, this analysis provides a stepwise analysis of MPI-ESM-MR`s capability and limitations in capturing the observed spring SST - summer EA link. As illustrated in Figure 4, the ensemble members of MR-30 exhibit high variability, albeit with correlations mostly failing to reach significance. In light of this variability, our approach aims to first utilise the complete set of ensemble members to construct a comprehensive understanding of the MR-30`s potential to reproduce the SST - EA link. In other words, this first approach aims at testing the use of the ensemble mean. The weakly positive SST - EA link at the 4-month lag found in MR-30 (beta = 0.04) differs from the link found in ERA-20C (beta = 0.22), highlighting that relying solely on the ensemble mean might not fully capture the capabilities of MR-30. We modified the manuscript in order to make our intention clearer and to stress that the weaker strength of the causal link found in MR-30 could be anticipated by our previous analysis. We report these changes in lines L268-285:

"The observed disparities between the model and observations, as highlighted in the spatial correlations and time series analyses depicted in Figs.4-5, prompt further investigation into the causal relationships within MR-30. To address this, we proceed to assess whether the model reproduces any of the observed causal links or presents alternative causal pathways. We construct three different CEN sets to evaluate, respectively, pi-Control, historical and initialised simulations with MR-30. The variables analysed in the CEN sets are SST, EA and SLP indices and the time lag of interest is spring - summer (3 and 4 months lag). While no causal links are found in the historical simulations, we find opposite causal links than those in ERA-20C for the pi-Control simulation, suggesting an atmospheric forcing from EA into the

extratropical North Atlantic (e.g. $\beta_{EA \to SST} \approx 0.22$), but no detected causal influence from the ocean on the atmosphere (Fig.6c).

Moving on to the initialised simulations, we leverage the entire 30-member ensemble of MR-30 to construct a comprehensive CEN spanning the full period (1908-2008), resulting in each constructed time series comprising 3030 years. We find that MR-30 is able to reproduce a weakly positive SST index - EA link (i.e. $\beta_{SST \to EA|SLP} = 0.04$) at 3-month lag (Fig.6a), but not at 4-month lag as detected in ERA-20C during the late period, and in much weaker strength (i.e. $\beta_{SST \to EA|SLP} = 0.22$, ERA20C). Moreover, we find a weak negative causal link from SST index to SLP index in the model (i.e. $\beta_{SST \to SLP |EA} = - 0.02$), as opposed to observations (i.e. $\beta_{SST \to SLP |EA} = 0.21$, Fig.3b). This finding aligns with Fig.5d-f, which shows that the area of positive correlations in MR-30 is displaced southwestwards with respect to ERA-20C. No causal links from SST index to EA or SLP indices are found when analysing only the late period (1958-2008). Next, we therefore investigate the causal link sensitivity to the sample size and focus on 45-year long timeseries covering the late period, allowing a direct comparison with the sensitivity analysis performed in ERA-20C (Fig.6b-d)."

**L232 There is no Fig. 6e**

Corrected.

**Figure 6 a and c. Node "EAP"→ EA.**

Modified as suggested.

**Figure 7a. It is not clear why the authors did not keep SLP-ind in the causal graph? Is there a reason why the colormap for the β -coef. is changed?**

We thank the reviewer for pointing this out. Upon reevaluation we decided to show in Fig.7a a causal graph containing SST index, SLP index and T2m_CE. The reason for that is that in Fig.7b we show a causal map for 3-month lag, and we want to provide a causal graph that is immediately comparable with it. This also answers your second question, why we changed the colormap for the causal graph in this panel: to make the comparison with Fig.7b easier.

**Discussion**
**P15LL288 For which time period?**

We rewrote the sentence as "Using ERA-20C, our CEN analysis confirms that the spring SST index proposed in Ossó et al. (2018) causally influences the variability of summer SLP in the Euro-Atlantic region with a 3-4 months delay during the late period (1958-2008)." (L343-344)

**Conclusions**

**P16 L323 "we find that"… but in L303 the authors state that the nonstationarity has been previously reported by other studies.**

An important finding of the present study is to find that the link between extratropical NA-SST (SST index) and summer EA is nonstationary. Rieke et al. 2021 finds that the link between tropical North Atlantic SSTs and summer EA (previously reported by Wullf et al. 2017) is nonstationary. To our knowledge, no other study has looked into the stationarity of extratropical NA-SST in particular.

**L326 Please, specify if this conclusion is for the reanalysis. Same for L330.**

We rewrote the sentences as "We find that this relationship in ERA-20C is only causal over the late period." and "In addition to summer EA, we find in ERA-20C that the spring SST index causally influences summer T2m (...)" (L383-385)

**L328 "that an external physical mechanism not included in our analysis" → the usage of PCMCI implies that the user makes an assumption that all variables are included to represent a specific mechanism. If user does assume that all variables are included in the causal graph/CEN, then LPCMCI should be rather used.**

Our assumption is that all variables relevant to the causality of the analysed physical mechanisms are indeed included in the analysis. Therefore, we chose to use PCMCI instead of LPCMCI. However, we do feel the need to point out, for transparency, that there could be other relevant variables that we cannot identify in the related literature, that may affect the results. This is true in virtually the entire body of studies that has applied PCMCI since 2014. Here, we want to stress that we analyse the causal relationship among the selected time series. Finally, although LPCMCI may results in the explicit information that (one) or more actors are missing, it would however no help us in identifying those variables in the physical set of fields.

**L333. Please, also summarize why the model does not reproduce the links from the reanalysis.**

We acknowledge the reviewer's observation; however, our analysis did not delve into the specific reasons for the model's inability to capture the causal links. A subsequent study would be necessary to address this comprehensively. Various factors could contribute to the low performance of MPI-ESM, including biases in the positioning of the jet stream (Pithan et al., 2016; Beverley et al., 2019), inaccurate representation of coupled ocean-atmosphere feedbacks (e.g. Ossó et al. 2020), and ensemble overdispersion stemming from initialisation

issues (Ho et al., 2013). To highlight the need for future work investigating the possible causes, we added in L397-399: "Exploring the causes behind the model's deficiencies in this aspect—such as whether they stem from inadequacies in representing crucial coupled ocean-atmosphere feedbacks or other processes—will be a significant focus for future research. "

**L334 "weakly"→ weak**

Modified as suggested.

**L341 "limited performance in reproducing"→ I suggest to highlight the reasons while giving such a statement.**

Please see the reply regarding L333 above.

**References**

Beverley, J. et al (2019). "The northern hemisphere circumglobal teleconnection in a seasonal forecast model and its relationship to European summer forecast skill." In: Climate dynamics 52.5-6, pp. 3759–3771.

Ho, C. et al (2013). "Examining reliability of seasonal to decadal sea surface temperature forecasts: The role of ensemble dispersion." In: Geophysical Research Letters 40.21, pp. 5770–5775

Pithan, F. et al (2016). "Climate model biases in jet streams, blocking and storm tracks resulting from missing orographic drag." In: Geophysical Research Letters 43.13, pp. 7231–7240

Ossó, A., Sutton, R., Shaffrey, L. and Dong, B., 2020. Development, amplification, and decay of Atlantic/European summer weather patterns linked to spring North Atlantic sea surface temperatures. Journal of Climate, 33(14), pp.5939-5951.

---

## Author Comment (AC2)

**Referee 2**

We appreciate the anonymous referee's valuable feedback, which helped identify weaknesses in the text. In response, we thoroughly revised the manuscript. We provide below a point-by-point response to each comment. Please note that the referee's comments are highlighted in **bold** font, while our answers are in regular font.

**Major comments:**

1) **Period. MA, AM, MAM? It is quite difficult to follow the various spring selections and to understand why one is chosen with respect to the other. As long as I understood, most results refer to AM, but the other seasons are cited here and there. I think this is quite confounding for the reader, and a single spring window would help, maybe just stating that changing it does not change the results.**

   We thank the reviewer for this comment. We choose to investigate both MA and MA spring windows to allow comparison with the study of Ossó et al. 2018 and 2020. In fact, we find it important to mention the different time means, since different causal links were detected in our analysis concerning AM or MA SST index (also, please see the reply concerning the specific comment to L166). This is nicely illustrated in Fig.3, where we find a causal link between MA SST index -> JA EA (Fig.3f, lag = 4) and between AM SST index -> JA SLP index (Fig.3c, lag= 3).

2) **The methods section 2.2 could be expanded to include a more detailed description of the bootstrap ensemble, which is currently just in the main text. It is unclear how the authors refer to the bootstrap ensemble throughout the text. Linked to this, the term "causal ensemble" or "causal timeseries" is not explained in the text. Also, since many indexes are considered in the text, a point-by-point list of all indexes considered with a unique and identifying name would help. For example the SST_Ridge just appears at the end of the results section, but would be useful to have it here for quick reference.**

   We thank the reviewer for these suggestions. We included two new sections and a table in the methodology (lines 169-188 in the revised manuscript):

   "2.4 Bootstrapping and ensemble subsampling

   We perform bootstrapping and ensemble subsampling (e.g. Dobrynin et al. (2018)) to investigate the sensitivity of the causal links to data sampling. When analysing 1958-2008 using observations in Sect.3.3, we randomly select 500 samples of 45-years long, i.e. excluding 6 random years at each iteration. Each of these time series

are then analysed with CEN using the same hyperparameters (see Sect.2.3). We perform a similar bootstrapping using MR-30, but additionally include a second step of ensemble subsampling. That is, we first randomly exclude 6 years from the analysed period. Next, we randomly select 1 ensemble member from the 30-member set. Each time series is then analysed with CEN. This process is repeated 2000 times. It is important to note that reducing the length of the time series in this way increases the variability and hence lowers the significance of the obtained β-coefficients. However, this should not by itself lower the strength of the coefficients themselves.

2.5 Predictive skill assessment

In Sect.3.4, we perform a predictive skill assessment for SLP, T2m and Z500 at lead times of 3-4 months in MR-30 against ERA-20C. For this assessment we use point-wise detrended anomaly correlation coefficient (ACC, Collins (2002)). We are interested in assessing the predictive skill conditioned to the strength of significant β-coefficients (p-value < 0.1). Our hypothesis is that the predictive skill in summer is likely to increase in cases where MR-30 is able to capture the causal link between spring SST index and summer EA, as opposed to cases where the model fails to capture the observed causal link. We refer to these time series as MR-30 bootstrap ensemble. For example, we shall assume that we are interested in calculating the conditioned predictive skill of JA Z500. To accomplish this task, we first identify the specific years and ensemble members that correspond to significant β-coefficients for the spring SST and summer EA. With this information, we can then sample JA Z500 to create a time series of similar length. In case more than one ensemble member is randomly selected in a given year, we calculate an ensemble mean. We then determine the ACC between the MR-30 bootstrap and ERA-20C."

We additionally included a table (Table 1) containing a list of the investigated variables. We also added in Sect. 2.2 (lines 124-128) "(...) In a second step, we analyse the impact of NA-SST on summer T2m using two additional indices, i.e. T2m_CE and T2m_Ridge (Sects. 3.2, 3.4). The T2mCE index is calculated as JA T2m anomalies averaged over the region 45∘N-55∘N; 10∘E-35∘E (indicated by a red box in Fig2f), and the T2mRidge index is calculated over the region 40°N-55°N; 15°W-34°W (indicated by a black box in Fig7b)."

3) **The end of the results section could possibly make for a separate subsection: see comment at lines 256-281. I think this is one of the most interesting results of the paper, but is currently difficult to grasp and could be easily lost in the main text.**

Following the reviewer's suggestion, we included a new section "3.5 Forecasts of opportunity: could causality help?" concerning these results.

4) **Physical pathway. The CEN framework is a powerful tool but should be used with caution. In particular two things are necessary, following Kretschmer et al. (2021):**
   a) **the CEN should only be used to "measure" a causal link for which there is already an hypothesized physical pathway;**
   b) **the causality of the link is always conditional to the choice of the variables included in the model, meaning that if a relevant variable is missing, the CEN result may be wrong.**
   **I think the authors respect both "requirements", but I suggest to:**
   c) **recall the possible physical mechanism behind the link when presenting the CEN. It is currently only cited in the introduction.**
   d) **put more emphasis and discuss on the possible impact of a missing process in the CEN.**

Thanks for your input. Concerning a), b) and d), we included a paragraph in Sect. 2.3 to provide a more in-depth discussion on the challenges of using CEN (please see below, under the reply to the specific comment for L131). Concerning c), we rewrote the beginning of Sect. 2.3 (L134-136) as "We use Causal Effect Network analysis (CEN, Runge et al. (2015); Kretschmer et al. (2016)) to test whether spring NA-SST anomalies causally influences the variability of summer SLP and temperature fields in the Euro-Atlantic sector during the 20th century, investigating the mechanism proposed in Ossó et al. (2018, 2020)."

**Specific comments**

**L94. The ocean state is derived  from?**

Text modified as suggested (L100).

**L94-97. It is not completely clear to me how the assimilation experiment is performed for the ocean. If is said at line 94 that an ocean-only simulation with MPI-OM forced with ERA-20C is performed. Is this ocean state then used for the nudging of the 30 ensemble members? I think this choice should be (quickly) motivated in the text.**

We thank the reviewer for raising this question. In the assimilation experiment for the ocean, an ocean-only simulation is conducted using the MPI-OM model, which is forced with ERA-20C atmospheric reanalysis data. This choice is made because it allows for a longer observational record compared to using an ocean reanalysis. Additionally, using ERA-20C for atmospheric forcing enables the production of century-scale assimilation runs without the

need to create a separate ocean reanalysis. The ocean state obtained from this ocean-only simulation is then used for nudging in the ensemble members. This means that the ocean model within the hindcasts is nudged towards the ocean state obtained from the ocean-only simulation forced with ERA-20C data. By using this approach, the initialization shock is reduced because the same ocean model is used in both the hindcasts and the assimilation run, ensuring consistency in model physics. A similar approach using MPI-ESM has been followed in Borchert et al. 2018.

To draw attention to this point, we added in Sect. 2.2 (L100-101): "To help reduce initialisation shock, the ocean state is derived from an ocean-only simulation performed with MPI-OM forced with the atmospheric variables from ERA-20C, thus maintaining consistency in model physics."

**L102. ..at lead times of 3-4 months..**

Text modified as suggested (L108).

**L106. I would say: "... the second principal component (PC) of the leading empirical orthogonal function (EOF) decomposition of ..."**

Text modified as suggested (L112).

**L131. Since the technique is pretty novel and can easily generate misunderstandings, I suggest the authors to add some further disclaimer for the reader. In particular, as implicitly stated a few lines above, a limitation of the Causal effect network analysis is that the choice of the variables to be considered is crucial for determining the causality of the link. In this sense, the possibility of a spurious correlation can never be completely excluded. I think this should be made clear in the text, expanding the sentence at L131.**

The reviewer has a very good point. We added the text below to convey this information (L144-154):

"We emphasise that the term "causal" should be interpreted cautiously within the context of this study. When we refer to causality, we mean causality relative to the set of investigated variables and under the specific assumptions considered in the PCMCI algorithm (such as the stationarity of time-series data). As a consequence, the possibility of spurious correlations cannot be entirely ruled out. The choice of variables included in the analysis is another crucial aspect for determining the causality of the identified links. Yet, this poses a challenge as including more variables enhances the credibility of causal discoveries but introduces complexities. For instance, accommodating numerous variables and significant time lags to address physical delays, like identifying atmospheric

teleconnections, leads to high dimensionality. This, in turn, can significantly affect the reliability of statistical outcomes. Hence, a successful application of CEN requires (such as for any data-driven method), expert knowledge of the underlying physical processes, including relevant variables, time-scales and temporal resolution. For a more detailed understanding of the CEN analysis and the PCMCI algorithm, we refer the reader to Runge (2018), which provides a comprehensive description of these techniques."

**Figure 1 caption. Specify the period considered for the SST anomaly. From the text it is spring SSTs and summer EA (L151), but this is not clear for the caption. Also, it is not clear what period panels c, d, e are referring to. In general, since different periods are considered for different variables, a suggestion would be to put the period as a subscript: SSTMA.**

Thank you for this comment. We rewrote the caption to highlight which period is being investigated, and included which months are included in the calculation of each index.

"Variability and linear relationships of EA in ERA-20C. a) Positive phase of the EA teleconnection, defined as the second EOF of July-August (JA) SLP. b) Regions used to calculate the NA-SST and SLP indices proposed in Ossó et al. (2018). c) Pointwise correlation of EA index with concurrent JA anomalies of 2-metre air temperatures in the full period (1908-2008). d) Same as c), for JA anomalies of total precipitation. e) Time series of April-May (AM) SST (blue) and JA EA (grey) indices in ERA-20C for 1908-2008, smoothed by a 3-year running mean. f) Running-correlation between AM SST and JA EA indices for a 20-year window. Coloured markers indicate significant correlations at the 95% confidence interval, illustrated by dashed lines."

**L151. Linked to the above. Is the spring SST referred to MAM, MA or AM?**

We rewrote the sentence as "A Pearson correlation analysis reveals a time-dependent relationship between the AM SST index and the EA in summer (Fig.1e)." (L195-196)

**L166. Why not use MAM? Is there some process changing significantly between early and late spring?**

We thank the reviewer for this question. First, we chose to analyse bimonthly means of SST to allow a direct comparison with the studies of Ossó (2018 and 2020). Second, we decided to include both March-April and April-May SST indices in the analysis because we found that in ERA-20C both indices can causally influence SLP in summer (Fig.3c,f).

**L200. The T2mCE looks like potentially correlated with EA with no lag (from the composite in Fig. 1).. isn't this correlation appearing in the CEN?**

We thank the reviewer for raising this point. The contemporaneous correlation does not appear in the CEN. The reason for that is two-fold. Firstly, we aim to identify precursor signals in spring, specifically before the initialisation of prediction systems typically in May, that could forecast the summer EA. Thus, we concentrated on uncovering interseasonal causal links, i.e. those between spring and summer. In other words, we used tau_min=3 and not tau_min=0. Secondly, we chose to perform our analysis using PCMCI (version 4.2 from Tigramite), which cannot identify causal links at lag 0.

**L207-8. I appreciate that the author acknowledge that something could be missing from the CEN. Also, could this mean that the observed causal link may be to some extent spurious, since some key process is missing from the CEN?**

We thank the reviewer for this question. Indeed, we acknowledge the concept of "causal sufficiency" and the possibility that some key processes may be absent from the CEN, potentially leading to spurious causal links. Adding more variables could potentially alter the network structure, highlighting the dynamic nature of causal inference within complex systems. For this reason, it's crucial to interpret the term "causal" cautiously within the context of our study. When we refer to causality, we do so relative to the set of investigated variables and the specific assumptions considered in the PCMCI algorithm. We recognise that the possibility of spurious correlations cannot be entirely ruled out, given the inherent challenges in variable selection and the complexities introduced by including additional variables.

However, we have good reasons to believe that the spring SST index -> summer EA link is unlikely to be spurious. A follow-up analysis including a tropical SST index (suggested as another predictor for summer EA, e.g. Wullf et al. 2017) in the network showed that the link is stable (see figure below).

[Figure]

Fig 1: Follow-up analysis showing a causal graph including the additional variable "SST_trop", following Wullf et al. 2017.

**L212. I wouldn't call this "skill in reproducing the summer EA" since Fig. 4a is evaluating the pdf of the EA index, so just from a statistical/climatological point of view.**

Indeed. We rewrote the sentence following the reviewer's suggestion as "We find that MPI-ESM generally captures the range of variability, although its performance in replicating the summer EA varies across different simulation sets." (L258-260).

**L216. In what sense is Fig. 4b showing "MR-30 capturing the temporal variability of the relationship in the early period" ? I think the only information is about the spread of the relationship, but I do not see a tendency for a negative correlation in the early period as observed for ERA20C.**

Thanks. We rewrote the sentence as "We find that the model shows limited skill, particularly in the late period." (L262-263)

**L221. I agree MR-30 looks slightly better for the early period, but still is quite far from ERA20C (the positive correlation in the southern North Atlantic is not significant and does not extend so much North). This is true for the ensemble mean, but have you checked whether some individual member is getting a response closer to the observed relation? This could possibly inform on the "missing" process in the chain.**

We thank the reviewer for raising this question. As you can see in Fig. 2 below, there is a great variability amongst ensemble members for correlations between AM SST index and JA SLP, with the majority showing too weak positive correlations, mostly displaced to the west, in comparison to Fig.4.c. This is not surprising for MR-30, which has shown limited performance in reproducing teleconnections in the North Atlantic European region, particularly in summer (e.g. Carvalho-Oliveira et al. 2022).

[Figure]

**Fig. 2** Correlations between AM SST index and JA SLP for each ensemble member in MR-30 in the full period (1908-2008).

**L222. I do not understand "first" in the sentence.**

Please note that this section has been rewritten, while addressing comments of the other reviewer. The new sentence is "The variables analysed in the CEN sets are SST, EA and SLP indices and the time lag of interest is spring - summer (3 and 4 months lag)." (L271-273).

**L228. 0.03 seems very small. Is it significant?**

Please note that this section has been rewritten, while addressing a specific comment of the other reviewer (concerning P9 L222-223). We unfortunately had used the wrong causal graph in Fig.6a by mistake, since over the course of our analysis we also tested the effect for $tau\_min = 2$. The correct causal graph containing the same set of variables but focusing on $tau\_min = 3$ and $tau\_max = 4$ is now in Fig. 6a. As a consequence, the causal link in question has a beta-coefficient of 0.04, and not 0.03. In both cases, the beta-coefficient is significant at the 0.1 level (also see L166 for the chosen CEN parameters).

**L245. What do you mean by "MR-30 causal timeseries"? If referred to the "MR-30 causal ensemble" in Fig. 6 caption, I would change the wording to something different.. e.g. MR-30 bootstrap ensemble**

We thank the reviewer for the suggestion. A new section in the methodology (2.5: Predictive skill assessment) seeks to clarify this point. Please note that this issue has been already addressed in the reply to "major comments" #2.

**L245. How do you perform the "predictive skill assessment"? By selecting random members with beta close to beta_1 and checking the skill only for those?**

We acknowledge that this section needed further explanation to improve readability. To conduct this predictive skill assessment, we first identify ensemble members with beta coefficients equal to beta_1, indicating significant relationships between the spring SST index and summer EA. We then assess the predictive skill of our target variable (in the case of Fig.6d, JA SLP) exclusively for these selected years and ensemble members. We hope that including Sect. 2.4 and 2.5 in the manuscript will clarify this step for the reader. We added, in particular "To accomplish this task, we first identify the specific years and ensemble members that correspond to significant β-coefficients for the spring SST and summer EA. With this information, we can then sample JA Z500 to create a time series of similar length. In case more than one ensemble member is randomly selected in a given year, we calculate

an ensemble mean. We then determine the ACC between the MR-30 bootstrap and ERA-20C." (L179-188)

**L249. How rare is this? The information might be relevant.**

About 1% of the times.

**L257. I can't easily see the contours in Fig. 7b. Would be better a separate figure, or black contours.**

Following the reviewer's suggestion, we splitted Fig.7 into two figures, so that the causal map in Fig. 7b can be seen more clearly. Please see Fig. 7 and Fig. 8 in the revised manuscript. We modified the beginning of the caption in Fig 7. as "Spatial features of the causal influence of spring SST index on summer climate. a) (...)" and in Fig. 8 "Does the spring SST index influence summer predictive skill in MR-30? a) (...)".

**L256-L281. I would separate this last part, since it is focussing on a different topic: how does the existence of a causal link between SST spring and T2m Ridge (a different predictand from the rest of the paper) influence the forecast skill on the Ridge region in the random MR-30 ensemble? Also, I had a hard time following the section, which I think should be separated from the rest, better framed and possibly expanded. I say this because the result looks interesting but is quite difficult to grasp from the current text. Also, the choice of the new predictand might look like a cherry pick, but I think it could be better motivated with the fact that it is the only causal link reproduced in the random MR-30 ensemble. The question "what happens to the skill when a causal link is reproduced?" seems relevant.**

Following the reviewer's suggestion, we included a new section "3.5 Forecasts of opportunity: could causality help?" to address these results. Also relevant for this section, we explained in more detail how the predictive skill assessment is achieved in the new Sect. "2.5 Predictive skill assessment". We also rewrote parts of the text to increase readability.

"We aim to identify a robust fingerprint of spring NA-SST on summer predictive skill, which could potentially enhance targeted forecasting opportunities (Mariotti et al., 2020). Our correlation analysis, as depicted in Fig. 2, indicates the potential influence of spring NA-SST on summer T2m variability across the Euro-Atlantic region during the late period. Thus, we conduct an additional causality analysis in ERA-20C to pinpoint the regions within the T2m field where a causal relationship with spring NA-SST is anticipated. We also explore whether this causal relationship might impact the predictive skill of MR-30." (L306-310)

**L312-316. This part looks a bit technical for the discussion, I suggest to remove it.**

Sentence removed as suggested.

**References**

Borchert, L.F., Müller, W.A. and Baehr, J., 2018. Atlantic ocean heat transport influences interannual-to-decadal surface temperature predictability in the North Atlantic region. *Journal of Climate*, *31*(17), pp.6763-6782.

Carvalho-Oliveira, J., Borchert, L.F., Zorita, E. and Baehr, J., 2022. Self-organizing maps identify windows of opportunity for seasonal European summer predictions. *Frontiers in Climate*, *4*, p.844634.

---

## Referee Report (RR1)

I thank the authors for considering my previous comments. While I find the results of this manuscript very interesting, I believe the authors ought to further improve the description and explanation of their results. For example, while NA-SST stands for North Atlantic sea surface temperatures, the authors defined the variable called "SST index" as a diagnostic of the SST variability over the North Atlantic region, which includes the anomalization and detrending of subtracted SSTs over two defined regions. Yet, in the discussion of the results, the authors frequently use simply "SST" or "NA-SST" while implying SST index (by the way, in the text of the manuscript it is SST index, while in Figures it is SST_index). The same issue persists for SLP and SLP index. I would highly appreciate it if the authors revised this issue. Maybe it is worth revising the naming of the variables used for the causal graphs as listed in Table 1? Perhaps there is no need to define an additional SST index and instead define NA-SST as the anomalized detrended monthly data of subtracted SSTs over two defined regions?

**Further issues**.

**Abstract.**

**L2**. The authors state that the analysis is done for 1908-2008. The next sentence in **L2-3** starts with "We find …" → this gives the impression that the results are related to the entire period. While in **L6-7** the authors clarify that this result is representative only for the 51-year period from 1958 to 2008. I recommend simplifying the sentence, e.g. "We find that in ERA-20C reanalysis the causal link from … to.. was evident/robust for the period 1958-2008 with an estimated causal effect expressed by … However, this causal connection was not evident/detected from the analysis of the entire period".

**L4.** In my opinion, **L4** reads very hard for an introduction; the reader is introduced to the NA-SST term, which further develops into the SST index. If the authors decide to keep both definitions, maybe the following makes it more clear: "NA-SST gradient in spring (expressed by SST index)…."

**L6.** "only evident during the **late** period from 1958 to 2008".

**L7**. How did the authors come up to a 45-year-long period? Why not randomly sampled 51 year-long period (like 1958-2008) to intercompare the strength of specific causal links based on timeseries of identical length?

**L11-15**. It is not clear which period of time the authors are talking about: full period (1908-2008) or a 51-year period (1958-2008)? Or all simulations have different analyzed period?

**L15**. Do authors mean "SST index"? Same in **L16**.

**Introduction**

**L28-30**. I recommend that this sentence is moved to ~L58, in order to keep the continuity of the introduction.

**L70**: The sentence in **L58** starts in a similar way. I recommend to unite the sentences from L28, L58 and L70 into one paragraph.

**Section 2.1.**

It is not clear from the text what the analysed periods for different MPI-ESM-MR simulations are (pre-industrial, historical, MR-30).

**Section 2.2.**

**L118.** by subtracting the average NA-SSTs or NA-SST anomalies?

**L120**. Isnt it the same as in L 106-107?

**L125**. Is it really just a second step? The first step was in **L111** to define a reference EA index. Then SST index seems to be ignored, and no "step" was given for it. And the second step is the analysis of the impact of NA-SST on temperature.

**L131**. I recommend the authors to be cautious here: NA-SST is not the index/variable as well as SLP. According to the authors notation SLP stands for the general definition of the sea level pressure, while SLP_ind represents the analysed variable. Therefore, the use of NA-SST definition here might also be also confusing.

**Section 2.3.**

**L136.** This part of the sentence could also perfectly fit to Abstract or Introduction.

**L140.** A note for the authors: they are partial correlations if the user chose partial correlation (ParCorr) as conditional independence test. I recommend to reformulate the sentence as follows: "This method/algorithm is based on iterative conditional independence testing among …"

**L168**. From the description it is not clear why did authors choose "parcorr"?

**Section 2.4.**

**L172.** Why is it necessary to exclude 6 random years from each iteration?

Side note: Tigramite has also the methodology for the bootstrap aggregation, see tutorial here https://github.com/jakobrunge/tigramite/blob/master/tutorials/causal_discovery/tigramite_tutorial_bootstrap_aggregation.ipynb

**Section 3.1**

**L191.** Large-scale cyclonic conditions where?

**Section 3.2**

**L195-196.** There is already the statement in Sect. 2.2 L117-118. Instead I suggest the following: "As explained in Sect.2.2, spring extratropical North Atlantic SSTs are represented via SST index, following the methodology of Osso et al.."

**L203** NA-SST or SST index? Here and below the authors use NA-SST. It is important to differentiate…

**L206-207** Repetition from L105-107.

**L207** "Correlation maps in Fig.2a-f"

**L208** summer SLP **index**?

**L213**. If there is further interesting material, it could be added to the Supplements.

**L214**. NA-SST anomalies or SST index?

**L217** Typo? No need of full stop.

**L219** Figure 2d-f depicts significant correlation between the AM SST….

Given the order of the explanation, I would move panels d-f after regression maps (and regression maps move up as panels d-f).

**L224-224**. The definition of the T2m$_{CE}$ is already provided in Section2.2 L126, however the coordinates of central Europe slightly differ. Please, correct this.

**L230**. Is it correct to state here "spring extratropical SST" or rather "SST index"?

**L233.** '… in ERA-20C , as defined is Sect. 2.1?2.2?' Please, make sure that methodology section clearly states the periods that were analysed by different simulations and includes the definition of "early" and "late" periods.

**L237**. Please, stay consistent: T2m$_{CE}$ (as in the text) or T2m_CE as in causal graph (Fig. 3a, Table 1)

**L239-240**. Shouldn't this information be stated in Sect. 2.3 instead of Results section?

**Fig 3.** Is there a typo in panel (b) with node "EAP"? It would be very useful if the authors would plot the nodes in panels (a) and (b) at the same position (implying that EA, SST_ind, and SLP_ind are plotted at the same position on both panels). This is easily done in Tigramite in plot_graph function using keyword **node_pos.**

**Caption Fig 3**. Circle → node? Typo "time tag" → "time lag"

**L243**. β$_{SST→EA}$--> β$_{SST\_ind→EA}$, same L245, L246, L253, L275.

**L246**. Can you really expect that the link SST → T2m$_{CE}$ is mediated via EA? It would be a mediator if the authors observe the link EA → T2m$_{CE}$. Contrary, there is no causal link from EA to T2m$_{CE}$, so how does EA mediate it?

**L248.** From the manuscript it follows that in order to avoid non-stationarity, the authors split the analysis to early and late periods. Yet in this sentence, the authors claim that since the full timeseries is nonstationary, they do not find a significant causal link in early period. It is not clear why they then found significant links in the late period.

**L250.** SLP index? Also in L256.

**L261**. Remove redundant full stop.

**L262.** How did authors find it out? Higher correlations?

**L263-264**. Do authors refer to just historical simulations?

**Figure 5.** There is no panels *g, h, i* as explained in the caption.

**L272.** From the text/Methodology it is not clear if pi-control, historical, and MR-30 are the same length as analysed ERA-20C data. Are the EA phases in agreement between the model and ERA-20C? If not, that all could contribute to the reasons why the direction of the link is different in comparison to the reanalysis.

**Section 3.5**

**L313.** I believe it should be β$_{SST\_ind}$ as well as in the caption of Fig.7 and L321, L322.

**L319.** This already explained in Sect 2.2. I recommend avoid duplication and rather refer the reader to Sect.2.2.

**L321.** ".. that are able to reproduce …"?

Conclusions.

L377. 1902→ 1908?

L387. "..detected in ERA-20C during the late period".

---

## Author Response (AR2)

Dear Editor,

We hope this message finds you well.

We have carefully revised the manuscript according to the helpful reviewer comments. You can find our responses as well as an updated manuscript attached, which we believe to be much improved due to your and their valuable work.

After further internal discussions regarding our manuscript, we recognised an opportunity to enhance the sensitivity analysis comparing the results from Causal Effect Networks (CEN) between observations and model outputs to address potential ambiguity present in some of the figures in our original manuscript. Specifically, in light of recent results published in Di Capua et al. (Weather and Climate Dynamics, 2023), we decided to improve our sensitivity analysis by fixing the causal parents based on those detected as significant for ERA-20C when calculating the causal graph for the late period in our MR-30 analysis. This adjustment allows us to provide a more equitable comparison between the strengths of the $\beta$-coefficients in the model and the observations and thereby further clarifies some of our original results. As a consequence, the corresponding figures have been enhanced accordingly. However, our central conclusions stand with the new, more thorough analysis.

We believe these revisions significantly enhance the clarity and robustness of our findings. Thank you for your consideration, and we look forward to your feedback.

Best regards,

Dr. Julianna Carvalho Oliveira

**Responses to reviewer R1**

**I thank the authors for considering my previous comments. While I find the results of this manuscript very interesting, I believe the authors ought to further improve the description and explanation of their results.**

We thank the reviewer for raising the need for further improving the explanation of our results. In the process of revising our manuscript accordingly, we have decided to enhance parts of the presented analysis by taking up a slightly different sensitivity analysis strategy as devised in a previous paper (Di Capua et al., Weather and Climate Dynamics, 2023) published after the original submission of our present manuscript (27 June 2023). The modified tests are described in detail in Sects. 2.4 and 3.5 of our revised manuscript and particularly address some results that could not be well explained in our original setting. We believe that these minor modifications of the manuscript content, though not having been explicitly requested by the reviewer, contribute substantially to further clarifying the reasons for some of our original findings on the sensitivity of causal links to the use of different datasets and time periods, respectively.

**For example, while NA-SST stands for North Atlantic sea surface temperatures, the authors defined the variable called "SST index" as a diagnostic of the SST variability over the North Atlantic region, which includes the anomalization and detrending of subtracted SSTs over two defined regions. Yet, in the discussion of the results, the authors frequently use simply "SST" or "NASST" while implying SST index (by the way, in the text of the manuscript it is SST index, while in Figures it is SST_index). The same issue persists for SLP and SLP index. I would highly appreciate it if the authors revised this issue. Maybe it is worth revising the naming of the variables used for the causal graphs as listed in Table 1? Perhaps there is no need to define an additional SST index and instead define NA-SST as the anomalized detrended monthly data of subtracted SSTs over two defined regions?**

We thank the reviewer for their thorough revision of our manuscript and appreciate the valuable feedback. We have carefully addressed all the specific points raised above. In particular, we have revised the manuscript to ensure consistent and accurate references to the indices and variables used when interpreting the results. Specifically, we have clarified the distinction between the SST and SST index (now consistently referred to as *SST index* throughout the text) and similarly for the SLP and SLP index. We also reviewed the variable names used in the figures to ensure consistency with the manuscript text. We are confident that these revisions resolve the issues highlighted.

**Further issues.**

**Abstract.**

**L2. The authors state that the analysis is done for 1908-2008. The next sentence in L2-3 starts with "We find …" → this gives the impression that the results are related to the entire period. While in L6- 7 the authors clarify that this result is representative only for the 51-year period from 1958 to 2008. I recommend simplifying the sentence,**

**e.g. "We find that in ERA-20C reanalysis the causal link from … to.. was evident/robust for the period 1958-2008 with an estimated causal effect expressed by … However, this causal connection was not evident/detected from the analysis of the entire period".**

We agreed and rephrased the sentence as: " *In the ERA-20C reanalysis, we find that the causal link from the meridional NA-SST gradient in spring (expressed by the SST index) to the summer EA is robust during the period from 1958 to 2008, with an estimated causal effect expressed by a beta-coefficient of about 0.2 (a 1 standard deviation change in the spring SST index causes a 0.2 standard deviation change in the EA 4 months later). However, this causal connection was not evident when analysing the entire period from 1908 to 2008.*"

**L4. In my opinion, L4 reads very hard for an introduction; the reader is introduced to the NA-SST term, which further develops into the SST index. If the authors decide to keep both definitions, maybe the following makes it more clear: "NA-SST gradient in spring (expressed by SST index)…."**

Please see the respective change in the response to L2-3.

**L6. ''only evident during the late period from 1958 to 2008''. L7. How did the authors come up to a 45-year-long period? Why not randomly sampled 51 year-long period (like 1958-2008) to intercompare the strength of specific causal links based on timeseries of identical length?**

Thank you for raising this point. In our analysis, we implement a leave-k-out cross-validation strategy, where we randomly exclude 6 years (approximately 12% of the dataset) at each iteration. This process generates 500 different samples of 45-year-long time series for the reanalysis data, allowing us to assess the robustness of the causal graph structure by testing how consistent the identified links are across various subsets of the data. Our method primarily serves as a sensitivity analysis of the causal graph. By excluding specific years while preserving the natural temporal order, we ensure the robustness of our findings.

**L11-15. It is not clear which period of time the authors are talking about: full period (1908-2008) or a 51-year period (1958-2008)? Or all simulations have different analyzed period?**

We performed a series of tests for the full period, 51-year periods (1908-1957 and 1958-2008) and the cross-validation of 45-year samples in the late period for all datasets. The only exception for this is shown in Fig.6c, where we show a causal graph for the pre-industrial simulation spanning the entire 1000 years. To account for this difference, we rewrote the sentence as: "*We then use different datasets with the MPI-ESM-MR to analyse the 1908-2008 period, focusing on a historical simulation and a 30-member initialised seasonal prediction ensemble. We specifically test the model's ability to reproduce the causal links detected in ERA-20C and evaluate its impact on the model's predictive skill for European summer climate. We find that MPI-ESM-MR generally fails to reproduce the causal link between the spring SST index and the summer EA across the datasets.*

*However, the 30-member initialised ensemble occasionally reproduces the causal link, though it typically underestimates its strength."*

**L15. Do authors mean "SST index"? Same in L16.**

This sentence has been modified and the new lines read as follows: *"We perform a predictive skill assessment conditioned on the spring SST index causal links for July-August sea level pressure, 500 hPa geopotential height and 2-metre air temperatures for predictions initialised in May. Our results suggest that while the overall impact may be limited, leveraging these causal links locally could help to constrain and modestly improve the seasonal prediction skill of European summer climate."*

**Introduction**

**L28-30. I recommend that this sentence is moved to ~L58, in order to keep the continuity of the introduction.**

We thank the reviewer for this suggestion. However, we believe that keeping the sentence about applying the PCMCI causal discovery algorithm at the end of the first paragraph of the introduction is important for immediately highlighting the core focus of our paper. Introducing the key method and its application early on helps readers quickly understand the primary aim of our study and its relevance.

**L70: The sentence in L58 starts in a similar way. I recommend to unite the sentences from L28, L58 and L70 into one paragraph.**

We thank the author for the suggestion. We altered the text as follows:
*"Nevertheless, while the linear regression-based analysis provided in Ossó et al. (2018) suggests a contribution of spring NASST to summer SLP variability, this approach does not imply causation. Disentangling the complex causal-effect pathways underlying the mechanism proposed in Ossó et al. (2020) over a long observational record is a crucial step in evaluating EA predictability in dynamical climate models. Although dynamical seasonal forecasts of European summer climate typically show limited skill (e.g., Mishra et al. (2019)), recent studies suggest that improving the representation of teleconnections can increase forecast skill (Oliveira et al., 2020; Carvalho-Oliveira et al., 2022; Schuhen et al., 2022). The physical mechanism connecting NA-SST variability and jet stream dynamics proposed in Ossó et al. (2020) provides a framework for assessing the broader influence of NA-SST on seasonal predictability of the EA, which is the aim of the present study.*

*In this paper, we use a Causal Effect Network based on PCMCI (hereafter CEN, Kretschmer et al. (2016)) to test the hypothesis that spring NA-SST causally drives a response in summer SLP and temperature fields in the Euro-Atlantic sector during 65 the 20th century. CEN overcomes spurious correlations caused by autocorrelation, indirect effects, or common drivers (Runge et al., 2014, 2019). It has been successfully applied to hypothesis testing for other tropical and mid-latitude teleconnections in the Atlantic-Pacific region (e.g., Karmouche et al. (2023)), the Indian Ocean (e.g., Di Capua et al. (2020a)), and the Arctic region (e.g., Siew et al. (2020); Kretschmer et al. (2020)).*

*Specifically, we use CEN to investigate the circumstances under which spring extratropical North Atlantic SSTs causally influence summer EA conditions and their associated impact on surface climate. We also analyse pre-industrial, historical, and initialised simulations with the Max Planck Institute Earth System Model in its mixed-resolution setup (MPI-ESM-MR, Dobrynin et al. (2018)) to evaluate model performance in reproducing the observed NA-SST–EA link, aiming to identify how this performance may constrain the seasonal prediction skill of European summer climate."*

**Section 2.1. It is not clear from the text what the analysed periods for different MPI-ESM-MR simulations are (pre-industrial, historical, MR-30).**

We see the issue and added the following paragraph at the end of Section 2.1: "*We focus our analysis on the 101-year period spanning 1908-2008, using data from both the historical simulation, the MR-30 hindcast ensemble, and ERA-20C. In addition, the piControl simulation, with its fixed external forcings, offers a unique opportunity to study long-term internal climate variability free from anthropogenic influences. To capture the full range of natural variability, we leverage the entire 1000-year period of the piControl in our analysis. This approach allows for benchmarking internal variability across observed, historical, and hindcast datasets.*"

**Section 2.2.**

**L118. by subtracting the average NA-SSTs or NA-SST anomalies?**

In the beginning of Section 2.2 we specify that we calculate anomalies for each variable. To make this clearer, we also rewrote the sentence as follows: " *As a second step, we calculate the SST index by subtracting the average NA-SST anomalies over the eastern box (35◦ W-20◦ W, 35◦ -42◦ N) from the average NA-SST anomalies over the western box (52◦ W-40◦ W, 42◦ -52◦ N), represented by green boxes in Fig.1b."*

**L120. Isnt it the same as in L 106-107? L125. Is it really just a second step? The first step was in L111 to define a reference EA index. Then SST index seems to be ignored, and no "step" was given for it. And the second step is the analysis of the impact of NA-SST on temperature.**

We improved the text structure taking this into account. We removed L120, since this was redundant given L106-107. Please see the reply to L118, where we include a second step for the calculation of the SST index. Lastly, we also rewrote L125 as follows: *"Next, we analyse the impact of NA-SST on summer T2m using two additional indices (...)"*

**L131. I recommend the authors to be cautious here: NA-SST is not the index/variable as well as SLP. According to the authors notation SLP stands for the general definition of the sea level pressure, while SLP_ind represents the analysed variable. Therefore, the use of NA-SST definition here might also be also confusing.**

We changed SLP to SLP index to make it clear.

**Section 2.3.**

**L136. This part of the sentence could also perfectly fit to Abstract or Introduction.**

We agree that the original phrasing had elements that could fit within the abstract or introduction. To address this, we revised the sentence to focus more directly on the application of the Causal Effect Network (CEN) method and removed the broader context. We rewrote the sentence as follows: *"We employ the Causal Effect Network (CEN) method (Runge et al., 2015; Kretschmer et al., 2016) to analyse the causal influence of the spring SST index on summer EA and temperature variability in the Euro-Atlantic sector."*

**L140. A note for the authors: they are partial correlations if the user chose partial correlation (ParCorr) as conditional independence test. I recommend to reformulate the sentence as follows: "This method/algorithm is based on iterative conditional independence testing among …"**

We agree with the reviewer's suggestion and rewrote the sentence as follows: *"This algorithm is based on iterative conditional independence testing amongst a set of time series (actors) to assess whether a link between a potential precursor and a target variable at a certain time lag is (...)"*

**L168. From the description it is not clear why did authors choose "parcorr"?**
**Section 2.4.**

We chose ParCorr for its suitability in detecting linear relationships, which aligns with our assumption regarding the SST index and EA. Its simplicity and interpretability, along with computational efficiency, make it ideal given our use of both observational and model data, as well as the need to perform cross-validation. We added this sentence for clarification: *"ParCorr was selected due to its ability to detect linear relationships, its computational efficiency, and its established application in related studies (e.g., Siew et al. 2020), providing clear and interpretable insights into conditional relationships between variables."*

**L172. Why is it necessary to exclude 6 random years from each iteration? Side note: Tigramite has also the methodology for the bootstrap aggregation, see tutorial here https://github.com/jakobrunge/tigramite/blob/master/tutorials/causal_discovery/tigramite_tutorial _bootstrap_aggregation.ipynb**

We appreciate the reviewer's comment regarding the exclusion of 6 random years, and thank the reviewer for sharing the tutorial. This decision was made to thoroughly assess the impact of sample size and interannual variability while preserving the chronological order of the remaining years. We opted against a bagging method, as it can lead to the repetition of the same years within a sample, which could complicate the interpretation of temporal dependencies essential to our analysis.

**Section 3.1**
**L191. Large-scale cyclonic conditions where?**

We rewrote the sentence for clarity as follows: *"The spatial pattern of the summer EA in its positive phase is characterised by large-scale cyclonic conditions across the Euro-Atlantic region, with the exception of the anticyclonic centre of action located south of Iceland and west of the British Isles."*

**Section 3.2**
**L195-196. There is already the statement in Sect. 2.2 L117-118. Instead I suggest the following: "As explained in Sect.2.2, spring extratropical North Atlantic SSTs are represented via SST index, following the methodology of Osso et al.."**

We rewrote the sentence as follows: *"As explained in Sect. 2.2, we evaluate spring extratropical North Atlantic SSTs via the SST index, following Ossó et al. 2018."*

**L203 NA-SST or SST index? Here and below the authors use NA-SST. It is important to differentiate…**

We rewrote the sentence as follows: *"This analysis suggests that the spring SST index - summer EA relationship is nonstationary."*

**L206-207 Repetition from L105-107.**

We deleted the sentence accordingly to avoid redundancy.

**L207 "Correlation maps in Fig.2a-f"**

Changed as suggested.

**L208 summer SLP index?**

The sentence correctly refers to SLP, as we investigate the point-by-point correlation of SST index with the SLP field, and not with the SLP index.

**L213. If there is further interesting material, it could be added to the Supplements.**

Thanks for this remark, we included the regression maps for early and late period as supplementary information.

**L214. NA-SST anomalies or SST index?**

We rewrote the sentence to improve readability as follows: *"Regression maps further suggest that the spring SST index is associated with summer NA-SST anomalies, which then influence atmospheric circulation (Fig.2g-i)."*

**L217 Typo? No need of full stop.**

We included the parenthesis missing.

**L219 Figure 2d-f depicts significant correlation between the AM SST…. Given the order of the explanation, I would move panels d-f after regression maps (and regression maps move up as panels d-f).**

We changed Fig. 2 as suggested.

**L224-224. The definition of the T2mCE is already provided in Section 2.2 L126, however the coordinates of central Europe slightly differ. Please, correct this.**

We corrected L126 and the coordinates are now the same in Table 1, L126 and L224: 46° N - 55° N; 11° E - 34° E.

**L230. Is it correct to state here "spring extratropical SST" or rather "SST index"?**

We rewrote the sentence to improve clarity as follows: *"To further test the robustness of the SST-EA relationship in ERA-20C, we evaluate whether spring SST index and summer EA are conditionally dependent. Specifically, we test the hypothesis that spring SST index is a causal driver for the summer EA, thus excluding autocorrelation effects or common drivers which could lead to spurious links."*

**L233. '… in ERA-20C , as defined is Sect. 2.1?2.2?' Please, make sure that methodology section clearly states the periods that were analysed by different simulations and includes the definition of "early" and "late" periods.**

Following the reviewer suggestion, we rewrote the sentence as follows: *"First, we build one CEN for each of the three investigated periods in ERA-20C, i.e. early, late and full periods, as defined in Sect.3.1."*

**L237. Please, stay consistent: T2mCE (as in the text) or T2m_CE as in causal graph (Fig. 3a, Table 1)**

As mentioned above, we made sure that all indices were written as subscripts in the manuscript to keep consistency, including those in Table 1.

**L239-240. Shouldn't this information be stated in Sect. 2.3 instead of Results section?**

Following the reviewer's suggestion, we moved this sentence to Sect. 2.3 and the last paragraph of Sect. 2.3 now reads: *"Lastly, the PCMCI parameters are chosen as follows: pc alpha = 0.2, alpha level to print results = 0.1, independence test = ParCorr, significance = 'analytic', masking type 'y'. ParCorr was chosen for its effectiveness in detecting linear relationships, computational efficiency, and established use in related studies (e.g. Siew et al. (2020)), offering clear insights into conditional relationships. Our CEN analysis focuses on τmin = 3 months and τmax = 4 months, which for simplicity we refer to as 3 and 4-month lags."*

**Fig 3. Is there a typo in panel (b) with node "EAP"? It would be very useful if the authors would plot the nodes in panels (a) and (b) at the same position (implying that EA, SST_ind, and SLP_ind are plotted at the same position on both panels). This is easily done in Tigramite in plot_graph function using keyword node_pos.**

We corrected the typo and now both nodes are correctly labeled as "EA".

**Caption Fig 3. Circle → node? Typo "time tag" → "time lag"**

Thanks for spotting the typo – changed as suggested.

**L243. $\beta_{SST}\rightarrow$ EA--> $\beta_{SST\_ind}\rightarrow$ EA, same L245, L246, L253, L275.**

Changed as suggested.

**L246. Can you really expect that the link SST $\rightarrow$ T2mCE is mediated via EA? It would be a mediator if the authors observe the link EA $\rightarrow$ T2mCE. Contrary, there is no causal link from EA to T2mCE , so how does EA mediate it?**

In the CEN analysis, we focus only on the 3-4 month lag, which is why no causal link is found from EA to $T2m_{CE}$, since the relationship between EA and $T2m_{CE}$ is contemporaneous. To answer the reviewer's question, we performed a mediation analysis using the path tracing rule. We found that while the SST index has a direct effect on $T2m_{CE}$ (-0.2) after accounting for EA, the indirect effect via EA is weaker and equal to -0.15. The direct effect of the SST index on $T2m_{CE}$, without conditioning on EA, is −0.31. Hence, the total effect equals −0.46, meaning that about 32.6% of the total effect of the SST index on $T2m_{CE}$ is mediated through EA. We rewrote the sentence as follows: *"We find a causal link of similar strength at a 3-month lag, $\beta_{SSTind}\rightarrow_{SLPind} \approx 0.21$ , between AM SST index and JA SLP index, as well as $\beta_{SSTind}\rightarrow_{T2mCE} \approx −0.2$ between AM SST index and JA $T2m_{CE}$. Using the path-tracing rule (e.g. Kretschmer et al. (2021)) we find that about a third of the influence from AM SST index on JA $T2m_{CE}$ is mediated via the EA."*

**L248. From the manuscript it follows that in order to avoid non-stationarity, the authors split the analysis to early and late periods. Yet in this sentence, the authors claim that since the full timeseries is nonstationary, they do not find a significant causal link in early period. It is not clear why they then found significant links in the late period.**

We agree that the sentence was misleading. We therefore rewrote it as follows: *"We find no significant causal links when using the early or full periods."*

**L250. SLP index? Also in L256.**

We rewrote the sentence as follows: *"Next, we test the sensitivity of the detected causal links between spring SST index and summer SLP to slight differences in the analysed years. We assess summer SLP using both EA and SLP indices."*

**L261. Remove redundant full stop.**

Changed accordingly.

**L262. How did authors find it out? Higher correlations?**

In the analysis of Fig. 4c, we compare the EA index calculated from MR-30 with the ERA-20C EA index. The fact that the observed EA lies within the range of MR-30 values, depicted as light grey dots, leads us to conclude that MR-30 mostly encompasses the observed variability.

**L263-264. Do authors refer to just historical simulations? Figure 5. There is no panels g, h, i as explained in the caption.**

We refer to both historical and MR-30. We rewrote the sentence to make it clearer. *"Next, we evaluate the model skill in reproducing the spring SST index - summer EA relationship. We find that both historical and MR-30 simulations show limited skill, particularly in the late period (Fig.4b, Fig5). A comparison between correlation maps computed for the evaluated periods shows that while historical simulations do not show agreement in the spatial pattern of the spring SST - summer EA relationship against observations (Fig.2a-c), the MR-30 ensemble mean shows an improvement in reproducing the mechanism (Fig.5d-f)."*
Additionally, we corrected the caption as indicated.

**L272. From the text/Methodology it is not clear if pi-control, historical, and MR-30 are the same length as analysed ERA-20C data. Are the EA phases in agreement between the model and ERA-20C? If not, that all could contribute to the reasons why the direction of the link is different in comparison to the reanalysis.**

We thank the reviewer for highlighting this point. Please refer to our response to the specific comment in Section 2.1 for details on the analysed periods. To ensure comparability, the EA indices for the model were calculated as projections onto the EA index derived from ERA-20C. This method allows for a consistent comparison of the EA phases between the model and ERA-20C.

**Section 3.5**
**L313. I believe it should be βSST_ind as well as in the caption of Fig.7 and L321, L322.**

Changed accordingly.

**L319. This already explained in Sect 2.2. I recommend avoid duplication and rather refer the reader to Sect.2.2.**

Changed as suggested.

**L321. ".. that are able to reproduce …"?**

Changed as suggested. The sentence now reads: *"Targeting the two causal regions CE and Ridge, we test the hypothesis that the predictive skill of summer surface (...)"*.

**L377. 1902→ 1908?**

Thank you for spotting the typo, we corrected it accordingly.

**L387. "..detected in ERA-20C during the late period".**

Changed as suggested.

**Responses to reviewer R2**

**The authors carefully revised the manuscript, rewriting significant parts, and satisfactorily addressed all issues raised in the first round. In particular, I find the new methods sections valuable for the reader's understanding and the whole paper easier to follow, despite the complex methodology. Also, section 3.5 is much clearer in this version and adds value to the overall story.**

We thank the anonymous reviewer for the positive feedback. We have carefully revised the manuscript, addressing all specific comments raised in the second round, as outlined below. During the revision, we decided to enhance parts of the analysis by adopting a slightly different sensitivity analysis strategy, inspired by a previous paper (Di Capua et al., Weather and Climate Dynamics, 2023), published after the original submission of our manuscript (27 June 2023). This updated analysis, detailed in Sections 2.4 and 3.5, significantly improves the clarity of our findings on the sensitivity of causal links across different datasets and time periods.

**I only have a couple of minor suggestions that could further improve the presentation:**

**- Major point 1: seasons. I understand now that MA and AM are considered to build on Ossò et al. (2018, 2020) and that this choice is relevant to the results (e.g. Fig. 3). I would state this clearly in the text at lines 106-107 (it is clear from your answer, I would put at least part of that in the main text).**

We appreciate the reviewer's comment. We included this information in the beginning of Sect. 2.2: *"We analyse bimonthly means in March-April (MA) and April-May (AM) for spring NA-SST and July-August (JA) SLP and T2m. We choose to investigate both MA and MA spring windows to allow comparison with previous studies (e.g. Ossó et al. 2018)."*

**- Robustness of the CEN. I find it interesting that the links identified by the CEN are robust to the introduction of a new driver (tropical SST following Wulff et al., 2017; Fig. 1 in your response), I would also state this in the text when commenting the possibility of other missing processes.**

Besides mentioning this link in the discussion, we now additionally wrote at the end of Sect. 3.2: *"This sensitivity in the causal link strength due to sampling suggests that the relationship between the spring SST index and summer SLP may be influenced by an external physical mechanism. Specifically, this could involve an additional variable not included in this CEN, such as the mechanism linking tropical SSTs described in Wulff et al. 2017."*

**- L303 (was 249). Please add the information on how rare are the positive causal links in MR-30 here, I think this might be relevant.**

Thank you for this remark. We included the percentage information and the new sentence reads as follows: *"Our sensitivity results suggest that the model predominantly fails to*

*reproduce the observed links between SST index and EA or SLP indices (Fig6b), showing only in about 5% of the cases beta-coefficients in the positive range as in ERA-20C (Fig.3)."*

---

## Author Response (AR3)

Dear Editor,

Thank you for accepting our manuscript. We have made the final adjustments as per your recommendations: specifically, we increased the font size in Figures 8 and 3b for better readability. Additionally, we have moved the three supplementary figures previously referenced as "SI" to Appendix A, following the guidelines for appendices. In case there is a preference to submit the figures as supplementary information instead, please let us know.

We appreciate your guidance throughout this process and look forward to the publication of our work in WCD. Please let us know if there are any additional steps needed.

Best regards,

Dr. Julianna Carvalho Oliveira

Research Fellow

Climate Causality Group

Universität Leipzig